# RE-EXAMINING LINEAR EMBEDDINGS FOR HIGH-DIMENSIONAL BAYESIAN OPTIMIZATION

## ABSTRACT

Bayesian optimization (BO) is a popular approach to optimize expensive-to-evaluate black-box functions. A significant challenge in BO is to scale to high-dimensional parameter spaces while retaining sample efficiency. A solution considered in previous literature is to embed the high-dimensional parameter space into a lower-dimensional manifold, often a random linear embedding. In this paper, we identify several crucial issues and misconceptions about the use of linear embeddings for BO. We thoroughly study and analyze the consequences of using linear embeddings and show that some of the design choices in current approaches adversely impact their performance. Based on this new theoretical understanding we propose ALEBO, a new algorithm for high-dimensional BO via linear embeddings that outperforms state-of-the-art methods on a range of problems, including learning a gait policy for robot locomotion.

## 1 INTRODUCTION

Bayesian optimization (BO) is a robust, sample-efficient technique for optimizing expensive-to-evaluate black-box functions (Mockus, 1989; Jones, 2001). BO has been successfully applied to diverse applications, ranging from automated machine learning (Snoek et al., 2012; Hutter et al., 2011) to robotics (Lizotte et al., 2007; Calandra et al., 2015; Rai et al., 2018). One of the most active topics of research in BO is how to extend current methods to higher-dimensional spaces. A common framework to tackle this problem is to consider a high-dimensional BO (HDBO) task as a standard BO problem in a low-dimensional embedding, where the embedding can be either linear (typically a random projection) or nonlinear (*e.g.* via a multi-layer neural network); see Sec. 2 for a full review. An advantage of this framework is to explicitly decouple the problem of finding low-dimensional representations suitable for optimization from the actual optimization technique.

In this paper we study the use of linear embeddings for HDBO, and in particular we re-examine prior efforts to use random linear projections. Random projections are attractive for BO because, by the Johnson-Lindenstrauss lemma, they can be approximately distance-preserving (Johnson & Lindenstrauss, 1984) without requiring any data to learn the embedding. Random embeddings come with several strong theoretical guarantees, but have shown mixed empirical performance for HDBO.

The contributions of this paper are: **1)** We provide new results that identify why linear embeddings have performed poorly in HDBO. We show that existing approaches produce representations that cannot be well-modeled by a Gaussian process (GP), or representations that likely do not contain an optimum (Sec. 4). **2)** We construct a representation with better properties for BO (Sec. 5): we improve modelability by deriving a Mahalanobis kernel tailored for linear embeddings and adding polytope bounds to the embedding, and we show how to maintain a high probability that the embedding contains an optimum. **3)** We show that using this representation for BO outperforms a wide range of previous approaches for HDBO, including on test functions up to $D = 1000$, and on real-world problems, such as gait optimization of a multi-legged robot (Sec. 6). These include the first results for HDBO with black-box constraints.

## 2 RELATED WORK

There are generally two approaches to extending BO into high dimensions. The first is to produce a low-dimensional embedding, do standard BO in this low-dimensional space, and then project up

to the original space for function evaluations. The foundational work on embeddings for BO is REMBO (Wang et al., 2016), which creates a linear embedding by generating a random projection matrix. Sec. 3 provides a thorough description of REMBO and several subsequent approaches based on random linear embeddings (Qian et al., 2016; Binois et al., 2019; Nayebi et al., 2019). If derivatives of $f$ are available, the active subspace method can be used to recover a linear embedding (Constantine et al., 2014; Eriksson et al., 2018), or approximate gradients can be used (Djolonga et al., 2013). BO can also be done in nonlinear embeddings through VAEs (Gómez-Bombarelli et al., 2018; Lu et al., 2018; Moriconi et al., 2019). An attractive aspect of random embeddings is that they can be extremely sample-efficient, since the only model to be estimated is a low-dimensional GP.

The second approach to extend BO to high dimensions is to make use of surrogate models that better handle high dimensions, typically by imposing additional structure on the problem. Work along these lines include GPs with an additive kernel (Kandasamy et al., 2015; Wang et al., 2017; Gardner et al., 2017; Wang et al., 2018; Rolland et al., 2018; Mutný & Krause, 2018), cylindrical kernels (Oh et al., 2018), or deep neural network kernels (Antonova et al., 2017). Random forest is used as the surrogate model in SMAC (Hutter et al., 2011). These methods produce trade-offs between sample efficiency of the model and the ability to effectively optimize the acquisition function.

Here, we focus on the embedding approach and in particular the use of linear embeddings for HDBO. Without box bounds, REMBO comes with a strong guarantee: with probability 1, the embedding contains an optimum (Wang et al., 2016, Thm. 2). However, if function evaluations are limited to the box bounds, as is typical in BO problems, REMBO requires a collection of heuristics for which there are no longer guarantees on performance. While REMBO can perform well in some HDBO tasks, subsequent papers have found it can perform poorly even on tasks with a true low-dimensional linear subspace (*e.g.* Nayebi et al., 2019). In this paper, we analyze the properties of linear embeddings as they relate to BO, and show how to improve the representation of the function we seek to optimize.

## 3 PROBLEM FRAMEWORK AND REMBO

In this section we define the problem framework and notation, and then describe BO via random linear projections (REMBO)—a promising method for HDBO—along with known challenges and follow-up work that has been proposed to address these issues.

**Bayesian optimization** We consider optimization problems of the form $\min_{\boldsymbol{x} \in \mathcal{B}} f(\boldsymbol{x})$ where $f$ is a black-box function and $\mathcal{B}$ are box bounds. We assume gradients of $f$ are unavailable. The box bounds on $\boldsymbol{x}$ specify the range of values that are reasonable or physically possible to evaluate. For instance, Gramacy et al. (2016) use BO for an environmental remediation problem in which each $x_i$ represents the pumping rate of a particular pump, which has physical limitations. The problem may also include nonlinear constraints $c_j(\boldsymbol{x}) \leq 0$ where each $c_j$ is itself a black-box function. BO is a form of sequential model-based optimization, where we construct a surrogate model for $f$ and use that model to identify which parameters $\boldsymbol{x}$ should be evaluated next, according to an explore-exploit strategy. The surrogate model is typically a GP, $f \sim \mathcal{GP}(m(\cdot), k(\cdot, \cdot))$, with mean function $m(\cdot)$ and a kernel $k(\cdot, \cdot)$. Under the GP prior, the posterior for the value of $f(\boldsymbol{x})$ at any point in the space is a normal distribution with closed-form mean and variance. Using that posterior, we construct an acquisition function $\alpha(\boldsymbol{x})$ that specifies the value of a function evaluation at $\boldsymbol{x}$, such as Expected Improvement (EI) (Jones et al., 1998). We find $\boldsymbol{x}^* \in \arg\max_{\boldsymbol{x} \in \mathcal{B}} \alpha(\boldsymbol{x})$, and evaluate $f(\boldsymbol{x}^*)$.

The GP is useful for BO because it provides a well-calibrated posterior in closed form. With typical kernels and acquisition functions, $\alpha(\boldsymbol{x})$ is differentiable and can be effectively optimized. However, with typical kernels like the ARD RBF kernel, there are significant limitations. GPs are known to predict poorly in high dimensions, which for a GP is $D$ larger than 15–20 (Wang et al., 2016; Li et al., 2016; Nayebi et al., 2019). This prevents BO from being a useful tool in high dimensions.

In HDBO, the objective $f : \mathbb{R}^D \to \mathbb{R}$ operates in a high-dimensional ($D$) space, which we call the *ambient space*. When using linear embeddings for HDBO, we assume there exists a low-dimensional linear subspace that captures all of the variation of $f$. Specifically, let $f_d : \mathbb{R}^d \to \mathbb{R}$, $d \ll D$, and let $\boldsymbol{T} \in \mathbb{R}^{d \times D}$ be a projection matrix from $D$ down to $d$ dimensions. The linear embedding assumption is that $f(\boldsymbol{x}) = f_d(\boldsymbol{T}\boldsymbol{x}) \ \forall \boldsymbol{x} \in \mathbb{R}^D$. $\boldsymbol{T}$ is unknown, and we only have access to $f$, not $f_d$. We assume without loss of generality that the box bounds are $\mathcal{B} = [-1, 1]^D$; the ambient space can always be scaled to these bounds.

**REMBO: Bayesian optimization via random embedding**  REMBO (Wang et al., 2016) generates a random projection matrix $\boldsymbol{A} \in \mathbb{R}^{D \times d_e}$ with each element drawn independently from $\mathcal{N}(0, 1)$ to specify a $d_e$-dimensional embedding. BO is done in the embedding to identify a point $\boldsymbol{y} \in \mathbb{R}^{d_e}$ to be evaluated, which is given objective value $f(\boldsymbol{A}\boldsymbol{y})$. The embedding dimension $d_e$ should satisfy $d_e \geq d$ for the REMBO guarantee of containing an optimum to hold.

The main challenges for using REMBO come when dealing with box bounds in the ambient space. We may select a point $\boldsymbol{y}$ in the embedding to be evaluated and find that its projection to the ambient space, $\boldsymbol{A}\boldsymbol{y}$, falls outside $\mathcal{B}$. The first challenge this poses is a theoretical challenge: $\mathbb{R}^{d_e}$ is guaranteed to contain an optimum, but that optimum is *not* guaranteed to project up to $\mathcal{B}$. When function evaluations are restricted to the box bounds, the embedding may not contain an optimum—it is not difficult to construct examples of this. REMBO has no theoretical guarantees in this setting. The second challenge posed by box bounds is the practical challenge of how function evaluations should be done for points that project up outside $\mathcal{B}$. Here REMBO introduces three heuristics. First, the embedding is given box bounds $[-\sqrt{d_e}, \sqrt{d_e}]^{d_e}$. BO will only select points within those bounds to be projected up and evaluated. Second, if a point $\boldsymbol{y}$ in the embedding projects up outside $\mathcal{B}$, then it is clipped to $\mathcal{B}$. Let $p_{\mathcal{B}} : \mathbb{R}^D \to \mathbb{R}^D$ be the $L^2$ projection that maps $\boldsymbol{x}$ to its nearest point in $\mathcal{B}$. A point $\boldsymbol{y}$ in the embedding is given objective value $f(p_{\mathcal{B}}(\boldsymbol{A}\boldsymbol{y}))$, which can always be evaluated. Note that clipping to $\mathcal{B}$ renders the projection of $\boldsymbol{y}$ to the ambient space a nonlinear transformation whenever $\boldsymbol{A}\boldsymbol{y} \notin \mathcal{B}$. Third, the optimization is done with $k$=4 separate projections, to improve the chances of generating an embedding that contains an optimum inside $[-\sqrt{d_e}, \sqrt{d_e}]^{d_e}$. Since these embeddings are independent, no data can be shared across them, which reduces sample efficiency.

**Extensions of REMBO**  Binois et al. (2015) consider the issue of non-injectivity, where the $L^2$ projection causes many points in the embedding to map to the same vertex of $\mathcal{B}$. They define a warped kernel that reduces non-injectivity, which is called REMBO-$\phi k_{\Psi}$. Binois et al. (2019) consider the issue of setting bounds on the embedding. They define a projection matrix $\boldsymbol{B} \in \mathbb{R}^{d \times D}$ that maps from the ambient space down to the embedding, and replace the $L^2$ projection with a projection $\gamma$ that maps $\boldsymbol{y}$ to the closest point in $\mathcal{B}$ that satisfies $\boldsymbol{B}\boldsymbol{x} = \boldsymbol{y}$. The $\gamma$ projection resolves the core challenge of REMBO related to setting bounds in the embedding: we can restrict the optimization in the embedding to points for which $\exists \boldsymbol{x} \in \mathcal{B}$ s.t. $\boldsymbol{B}\boldsymbol{x} = \boldsymbol{y}$, and so heuristic box bounds in the embedding are no longer required. The $\gamma$ projection projects to the same points on the facets of $\mathcal{B}$ as the $L^2$ projection. Paired with the warped kernel of Binois et al. (2015), this is called REMBO-$\gamma k_{\Psi}$.

Binois (2015) studies different choices for the projection matrix and shows that BO performance can be improved for small $d$ by sampling each row of $\boldsymbol{A}$ from the unit hypersphere $\mathbb{S}^{d_e-1}$. If $\boldsymbol{z} \sim \mathcal{N}(\boldsymbol{0}, \boldsymbol{I}_{d_e})$, then $\frac{\boldsymbol{z}}{||\boldsymbol{z}||}$ is a random sample from $\mathbb{S}^{d_e-1}$, so this amounts to normalizing the rows of the usual REMBO projection matrix.

HeSBO (Nayebi et al., 2019) is a recent extension of REMBO that avoids clipping to $\mathcal{B}$ and heuristic box bounds in the embedding by changing the projection matrix $\boldsymbol{A}$. In $d_e = 1$, it is easy to see that the projection matrix $\boldsymbol{A} = \boldsymbol{1}$, which sets every $x_i = y$, is optimal. With this projection we can set bounds of $[-1, 1]$ on the embedding and there is no need for $L^2$ projections because every point in the embedding will map to a point in $\mathcal{B}$. HeSBO extends this to $d_e > 1$ by setting each row of $\boldsymbol{A}$ to have a single non-zero element, which is randomly set to $\pm 1$. The column with the non-zero value is chosen uniformly at random. Thus, each parameter in the ambient space is mapped directly to a parameter in the embedding: $x_i = \pm y_j$, where $j$ is sampled uniformly from $\{1, \ldots, d_e\}$ and $\pm$ is chosen uniformly at random. The embedding is given box bounds of $[-1, 1]^{d_e}$.

## 4 Challenges with Linear Embeddings

Heuristics for handling box bounds when utilizing linear embeddings introduce several issues that impact HDBO performance. We highlight one recent observation from Binois et al. (2019), that most points in the embedding project up outside the box bounds, and discuss three novel observations about how existing methods can make it difficult to learn high-dimensional surrogates.

**Projection to the facets of $\mathcal{B}$ produces a nonlinear distortion in the function.**  The function value at any point in the embedding is measured as $f(p_{\mathcal{B}}(\boldsymbol{A}\boldsymbol{y}))$. For points $\boldsymbol{y}$ that project up outside of $\mathcal{B}$, this will be a nonlinear mapping from the embedding to the ambient space, despite the use of a

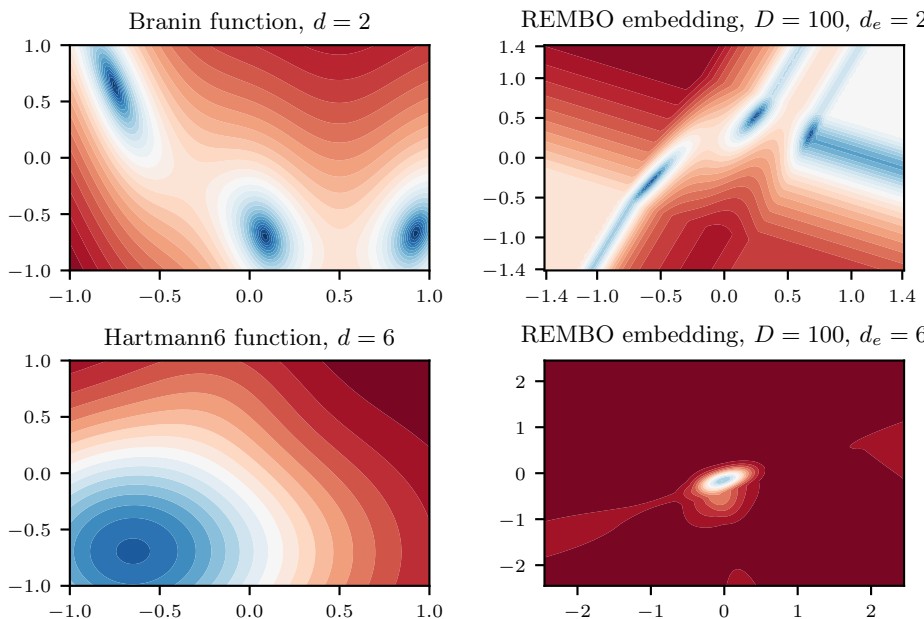

Figure 1:   A visualization of REMBO embeddings for two test functions. *(Top left)* The Branin function, $d$=2, extended to $D$=100. *(Top right)* A REMBO embedding of the $D$=100 Branin function. *(Bottom left)* A center slice of the $d$=6 Hartmann6 function, similarly extended to $D$=100. *(Bottom right)* The same slice of a REMBO embedding of that function. The embedding produces distortions in the function that render it difficult to model.

linear embedding. This has a powerful, detrimental effect on the ability to model $f$ in the embedding. Fig. 1 provides visualizations of an actual REMBO embedding for two classic test functions: the Branin ($d$=2) and Hartmann6 ($d$=6) functions, both extended to $D$=100 by adding unused variables. The REMBO embedding for the Branin function contains all three optima, however there is visible distortion to the function caused by the the clipping to $\mathcal{B}$. The embedding for the Hartmann6 function is even more heavily distorted.

Even if the function is well-modeled by a GP in the true low-dimensional space, the distortion produced by the REMBO projection transforms it into one on the embedding that is not appropriate for a GP. This can happen for any embedding strategy that cannot guarantee all points in the embedding project into $\mathcal{B}$. The distortion induced by mapping to the facet depends on the relative angles of the facet and the true embedding. Projection to a facet essentially induces a non-stationarity in the kernel: each of the $2D$ facets sits at different angles to the true subspace, and so the change in the rate of function variance will differ for each. To correct for the non-stationarity, we would have to estimate the true subspace $\boldsymbol{T}$, which with $d \times D$ entries is not feasible for $D$ large.

The idea behind using low-dimensional embeddings for HDBO is that it enables the use of standard BO techniques on the embedding. However, from these results we see that for the REMBO projection with box bounds we cannot expect to successfully model the function on the embedding with a regular GP. The problem is especially acute for $d_e > 2$ where, as we will see next, nearly all points in the embedding map to one of the $2D$ facets.

**Most points in the embedding map to the facets of $\mathcal{B}$.**   Fig. 2 shows the probability that an interior point in the embedding projects up to the interior of $\mathcal{B}$. This is measured empirically by sampling $\boldsymbol{y}$ uniformly at random from $[-\sqrt{d_e}, \sqrt{d_e}]^{d_e}$, sampling $\boldsymbol{A}$ with $\mathcal{N}(0,1)$ entries, and then checking if $\boldsymbol{Ay} \in \mathcal{B}$ (with 1000 samples). Even for small $D$, with $d_e > 2$ practically all of the volume in the embedding projects up outside the box bounds, and is thus clipped to a facet of $\mathcal{B}$.

This is an issue because it means the optimization will be done primarily on the facets of $\mathcal{B}$ and not in the interior, which will likely not even be reached in a typical BO initialization. We saw in Fig. 1 that the function behaves very differently on points projected to the facets, and that these parts of

the space can be hard to model with a GP. The problem cannot be resolved by simply shrinking the box bounds in the embedding. Binois et al. (2019) provide an excellent study of the issue of setting bounds in the embedding and show that with the REMBO strategy there is no good way to do this. The projection of $\mathcal{B}$ onto the embedding produces a star-shaped object called a zonotope, which has up to $2 \sum_{i=0}^{d-1} \binom{D-1}{i}$ vertices (Ferrez et al., 2005). Shrinking box bounds in the embedding cuts off the vertices of the zonotope and increases the chance of not containing an optimum.

**Linear projections do not preserve product kernels.** Although less visible than that produced by the projection to the facets, there is also distortion to interior points just from the linear projection $\boldsymbol{A}$. The ARD kernels typically used in GP modeling are product kernels that decompose the covariance into the covariance across each dimension. Inside the embedding, moving along a single dimension will move across all dimensions of the ambient space, at rates depending on the projection matrix. Consider moving along a single dimension in the embedding, from $\boldsymbol{y}_1$ to $\boldsymbol{y}_2$ where only a single element has changed. The corresponding points in the ambient space are $\boldsymbol{x}_1 = \boldsymbol{A}\boldsymbol{y}_1$ and $\boldsymbol{x}_2 = \boldsymbol{A}\boldsymbol{y}_2$: even though $\boldsymbol{y}_1$ and $\boldsymbol{y}_2$ differ in only one element, $\boldsymbol{x}_1$ and $\boldsymbol{x}_2$ will differ in all their elements. Thus a product kernel in the true subspace will not produce a product kernel in the embedding; this is shown mathematically in Proposition 1.

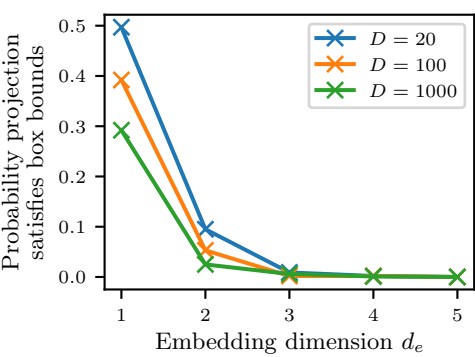

Figure 2: The probability that a randomly selected point in the REMBO embedding satisfies the ambient box bounds after being projected up. For $d_e > 2$, nearly all points in the embedding map outside the box bounds.

**Linear embeddings can have a low probability of containing an optimum.** HeSBO avoids the challenges of REMBO related to box bounds: all interior points in the embedding map to interior points of $\mathcal{B}$, and there is no need for the $L^2$ projection and thus the ability to model in the embedding is improved. However, for $d_e > 1$ there is no guarantee that the embedding will contain an optimum, and in fact the probability of containing an optimum can be quite low. Consider the example of an axis-aligned true subspace: $f$ operates only on some set of $d$ elements of $\boldsymbol{x}$, which we denote $\mathcal{I} = \{i_1, \ldots, i_d\}$. For $d = 2$ and $d_e \geq 2$, there are three possible embeddings: $x_{i_1}$ and $x_{i_2}$ map to different features in the embedding, $x_{i_1} = x_{i_2}$, or $x_{i_1} = -x_{i_2}$. These three embeddings are visualized in Appendix A.1. In the first case the embedding successfully captures the entire true subspace and we can expect the optimization to be successful. However, in the other two cases the embedding is only able to reach the diagonals of the true subspace, which, unless $f$ happens to have an optimum on the diagonal, will not reach the optimal value. Under a uniform prior on the location of optima, we can compute analytically the probability that the HeSBO embedding contains an optimum (see Appendix A.1). The probability is independent of $D$, but is low for even moderate values of $d$. For instance, with $d = 6$, $d_e = 20$ gives only a 44% chance of recovering an optimum.

Relative to REMBO, HeSBO improves the ability to effectively model and optimize in the embedding, but reduces the likelihood of the embedding containing an optimum. Empirically, this trade-off leads to HeSBO having better BO performance than REMBO. Like HeSBO, here we wish to eliminate the $L^2$ projection and thus improve our ability to model and optimize in the embedding. We will show that this can be done while maintaining a much higher chance of the embedding containing an optimum, which will further improve BO performance.

## 5 LEARNING AND OPTIMIZING IN LINEAR EMBEDDINGS

We now show how to overcome the embedding issues described in Sec. 4. Similarly to Binois et al. (2019), we define the embedding via a matrix $\boldsymbol{B} \in \mathbb{R}^{d_e \times D}$ that projects from the ambient space down to the embedding, and $f_B(\boldsymbol{y}) = f(\boldsymbol{B}^\dagger \boldsymbol{y})$ as the function evaluated on the embedding, where $\boldsymbol{B}^\dagger$ denotes the matrix pseudo-inverse. The new techniques we develop here are applicable to any linear embedding, not just random embeddings.

## 5.1 A KERNEL FOR LEARNING IN A LINEAR EMBEDDING

As discussed in Sec. 4, a product kernel over dimensions of the true subspace (ARD) does not translate to a product kernel over dimensions in the embedding. However, stationarity in the true subspace does imply stationarity in the embedding, and this result gives the appropriate kernel structure.

**Proposition 1.** *Suppose the function on the true subspace is drawn from a GP with an ARD RBF kernel: $f_d \sim \mathcal{GP}(m(\cdot), k_{RBF}(\cdot, \cdot))$. For any pair of points in the embedding $\boldsymbol{y}$ and $\boldsymbol{y}'$,*

$$Cov[f_B(\boldsymbol{y}), f_B(\boldsymbol{y}')] = \sigma^2 \exp\left(-(\boldsymbol{y} - \boldsymbol{y}')^\top \boldsymbol{\Gamma}(\boldsymbol{y} - \boldsymbol{y}')\right) ,$$

*where $\sigma^2$ is the kernel variance of $f_d$, and $\boldsymbol{\Gamma} \in \mathbb{R}^{d_e \times d_e}$ is symmetric and positive definite.*

*Proof.* To determine the covariance in function values of points in the embedding, we first project up to the ambient space and then project down to the true subspace

$$f_B(\boldsymbol{y}) = f(\boldsymbol{B}^\dagger \boldsymbol{y}) = f_d(\boldsymbol{T}\boldsymbol{B}^\dagger \boldsymbol{y}) .$$

Then,

$$\begin{aligned}
\mathrm{Cov}[f_B(\boldsymbol{y}), f_B(\boldsymbol{y}')] &= \mathrm{Cov}[f_d(\boldsymbol{T}\boldsymbol{B}^\dagger \boldsymbol{y}), f_d(\boldsymbol{T}\boldsymbol{B}^\dagger \boldsymbol{y})] \\
&= \sigma^2 \exp\left(-(\boldsymbol{T}\boldsymbol{B}^\dagger \boldsymbol{y} - \boldsymbol{T}\boldsymbol{B}^\dagger \boldsymbol{y}')^\top \boldsymbol{D}(\boldsymbol{T}\boldsymbol{B}^\dagger \boldsymbol{y} - \boldsymbol{T}\boldsymbol{B}^\dagger \boldsymbol{y}')\right) ,
\end{aligned}$$

where $\boldsymbol{D} = \mathrm{diag}\left(\left[\frac{1}{2\ell_1^2}, \dots, \frac{1}{2\ell_d^2}\right]\right)$. Let $\boldsymbol{\Gamma} = (\boldsymbol{T}\boldsymbol{B}^\dagger)^\top \boldsymbol{D}(\boldsymbol{T}\boldsymbol{B}^\dagger)$. Because $\boldsymbol{D}$ is positive definite, it follows that $\boldsymbol{\Gamma}$ is symmetric and positive definite. $\square$

This kernel replaces the ARD Euclidean distance with a Mahalanobis distance, and so we refer to it as the Mahalanobis kernel. Similar kernels have been used for GP regression in other settings (Vivarelli & Williams, 1999; Snelson & Ghahramani, 2006). This result shows that the impact of the linear projection on the kernel can be correctly handled by fitting a $\frac{d_e(d_e+1)}{2}$-parameter distance metric rather than the typical $d_e$-parameter ARD metric. The use of this kernel is vital for obtaining good model fits in the embedding. Appendix A.2 shows GP predictive performance on a linear embedding of the Hartmann6 function, in which an ARD RBF kernel entirely fails to predict, while the Mahalanobis kernel does not. We handle uncertainty in $\boldsymbol{\Gamma}$ by posterior sampling from a Laplace approximation of its posterior; this is described in the appendix.

## 5.2 AVOIDING NONLINEAR PROJECTIONS

The most significant distortions seen in Fig. 1 result from clipping projected points to $\mathcal{B}$. We can avoid this by constraining the optimization in the embedding to points that do not project up outside the bounds, that is, $\boldsymbol{B}^\dagger \boldsymbol{y} \in \mathcal{B}$. Let $\alpha(\boldsymbol{y})$ be the acquisition function evaluated in the embedding that we wish to optimize. We select the next point to evaluate by solving

$$\max_{\boldsymbol{y} \in \mathbb{R}^{d_e}} \alpha(\boldsymbol{y}) \quad \text{subject to} \quad -\boldsymbol{1} \leq \boldsymbol{B}^\dagger \boldsymbol{y} \leq \boldsymbol{1} . \tag{1}$$

Note that there are no box bounds on the embedding. The constraints $-\boldsymbol{1} \leq \boldsymbol{B}^\dagger \boldsymbol{y} \leq \boldsymbol{1}$ form a polytope, which is convex and can be efficiently optimized over with off-the-shelf optimization tools. Appendix A.3 provides visualizations of the embedding subject to these constraints. Within this space, the projection is entirely linear and can be effectively modeled with the GP described in Sec. 5.1.

## 5.3 THE PROBABILITY THE EMBEDDING CONTAINS AN OPTIMUM

Restricting the embedding with the constraints in (1) eliminates distortions from clipping to $\mathcal{B}$, but it also reduces the volume of the ambient space that can be reached from the embedding and thus reduces the probability that the embedding contains an optimum. To understand the performance of BO in the linear embedding, it is critical to understand this probability, which we denote $P_{\text{opt}}$. Recall that even with clipping, the REMBO theoretical result does not hold when function evaluations are restricted to box bounds, and so even REMBO will generally have $P_{\text{opt}} < 1$.

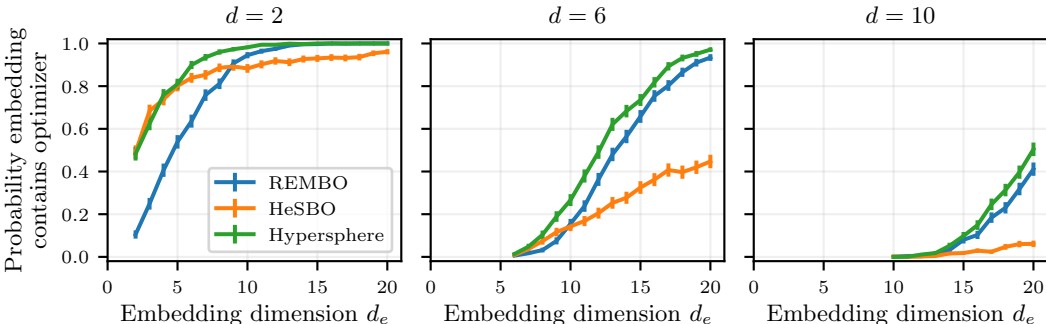

Figure 3: Probability the embedding contains an optimum ($P_{\text{opt}}$) when restricted to the constraints of (1), under a uniform prior for the location of the optima and $D = 100$, for three embedding strategies. Setting $d_e > d$ rapidly increases $P_{\text{opt}}$, and high probabilities can achieved with reasonable values of $d_e$. Hypersphere sampling produces the best embedding, particularly for $d$ small.

$P_{\text{opt}}$ depends on where the optima are in the ambient space—for instance, an optimum at $\mathbf{0}$ will always be contained in the embedding. Suppose the true subspace has an optimum at $\boldsymbol{z}^*$. Then, $\mathcal{O}(\boldsymbol{T}, \boldsymbol{z}^*) = \{\boldsymbol{x} : \boldsymbol{T}\boldsymbol{x} = \boldsymbol{z}^*\}$ defines the set of optima in the ambient space. We wish to determine if any of these optima can be reached from the embedding. The points $\boldsymbol{x}$ that can be reached from the embedding are those for which there exists a $\boldsymbol{y}$ in the embedding that projects up to $\boldsymbol{x}$, that is, $\boldsymbol{B}^\dagger \boldsymbol{y} = \boldsymbol{x}$. Since the embedding itself is produced from the projection $\boldsymbol{B}\boldsymbol{x}$, $\mathcal{E}(\boldsymbol{B}) = \{\boldsymbol{x} : \boldsymbol{B}^\dagger \boldsymbol{B}\boldsymbol{x} = \boldsymbol{x}\}$ defines the set of points in ambient space that can be reached from the embedding. The embedding contains an optimum if and only if the intersection $\mathcal{O}(\boldsymbol{T}, \boldsymbol{z}^*) \cap \mathcal{E}(\boldsymbol{B}) \cap \mathcal{B}$ is non-empty. Given a prior for the locations of optima (that is, over $\boldsymbol{T}$ and $\boldsymbol{z}^*$), we can compute $P_{\text{opt}}$ as

$$P_{\text{opt}} = \mathbb{E}_{\boldsymbol{B},\boldsymbol{T},\boldsymbol{z}^*} \left[ \mathbf{1}_{\mathcal{O}(\boldsymbol{T},\boldsymbol{z}^*)\cap\mathcal{E}(\boldsymbol{B})\cap\mathcal{B}\neq\varnothing} \right] . \tag{2}$$

Importantly, $\mathcal{O}(\boldsymbol{T}, \boldsymbol{z}^*)$, $\mathcal{E}(\boldsymbol{B})$, and $\mathcal{B}$ are all polyhedra, so their intersection can be tested by solving a linear program (see Appendix A.4). The expectation can be estimated with Monte Carlo sampling from the prior over $\boldsymbol{T}$ and $\boldsymbol{z}^*$ and from the chosen generating distribution of $\boldsymbol{B}$.

For our analysis here, we give $\boldsymbol{T}$ a uniform prior over axis-aligned subspaces as described in Sec. 4, and we give $\boldsymbol{z}^*$ a uniform prior in that subspace. Under these uniform priors, we can evaluate (2) to compute $P_{\text{opt}}$ as a function of $\boldsymbol{B}$, $D$, $d$, and $d_e$. Fig. 3 shows these probabilities for $D = 100$ as a function of $d$ and $d_e$, for three strategies for generating the projection matrix: the REMBO strategy of $\mathcal{N}(0, 1)$, the HeSBO projection matrix, and the unit hypersphere sampling described in Sec. 4. Increasing $d_e$ above $d$ rapidly improves the probability of containing an optimum. For $d = 6$, with $d_e = 6$ the probability is nearly 0, while increasing $d_e$ to 12 is sufficient to raise it to 0.5 and with $d_e = 20$ it is nearly 1. Across all values of $d$ and $d_e$, hypersphere sampling produces the embedding with the best chance of containing an optimum. Appendix A.4 shows $P_{\text{opt}}$ for more values of $D$ and $d$. By using hypersphere sampling and selecting $d_e > d$, we can maintain a high $P_{\text{opt}}$ while still avoiding clipping to $\mathcal{B}$.

### 5.4 A New Method for BO with Linear Embeddings: ALEBO

We combine the results and insight gained into a new method for HDBO, which we call adaptive linear embedding BO (ALEBO), since the kernel metric and embedding bounds are adapted with the choice of $\boldsymbol{B}$. The approach is given in algorithm form in Algorithm 1. Code is available at github.com/anonymized-for-review. In Line 1 the embedding is specified by generating a random projection matrix. We use hypersphere sampling, which gave the best $P_{\text{opt}}$ in Fig. 3 among strategies tried here, but this could be replaced with a different projection strategy should one be more appropriate for a particular setting.

## 6 Benchmark Experiments

We evaluate the performance of ALEBO on synthetic HDBO tasks, and compare its performance to a broad selection of HDBO methods. We include in these benchmarks: REMBO and HeSBO;

---

**Algorithm 1:** ALEBO method for high-dimensional BO in a linear embedding.

**Data:** $D$, $d_e$, $n_{\text{init}}$, $n_{\text{BO}}$.
**Result:** Approximate optimizer $\boldsymbol{x}^*$.

1 Generate a random projection matrix $\boldsymbol{B}$ by sampling $D$ points from the hypersphere $\mathbb{S}^{d_e-1}$.
2 Generate $n_{\text{init}}$ random points $\boldsymbol{y}^i$ in the embedding using rejection sampling to satisfy polytope (1).
3 Let $\mathcal{D} = \{(\boldsymbol{y}^i, f(\boldsymbol{B}^\dagger \boldsymbol{y}^i))\}_{i=1}^{n_{\text{init}}}$ be the initial data.
4 **for** $j = 1, \ldots, n_{BO}$ **do**
5     Fit a GP by maximizing marginal log-likelihood of $\mathcal{D}$, with the Mahalanobis kernel.
6     Draw posterior samples of $\boldsymbol{\Gamma}$ using a Laplace approximation. Marginalize over the posterior
      with moment matching.
7     Use the GP to find $\boldsymbol{y}^j$ that maximizes the acquisition function according to (1).
8     Update $\mathcal{D}$ with $(\boldsymbol{y}^j, f(\boldsymbol{B}^\dagger \boldsymbol{y}^j))$
9 **return** $\boldsymbol{B}^\dagger \boldsymbol{y}^*$, *for the best point* $\boldsymbol{y}^*$.

---

REMBO variants $\phi k_\Psi$ (Binois et al., 2015) and $\gamma k_\Psi$ (Binois et al., 2019); additive kernel methods Add-GP-UCB (Kandasamy et al., 2015) and Ensemble BO (EBO) (Wang et al., 2018); SMAC, which uses a random forest model; CMA-ES, an evolutionary strategy (Hansen et al., 2003); and quasirandom search (Sobol). For ALEBO we took $d_e = 2d$ for these experiments. In their evaluation of HeSBO, Nayebi et al. (2019) used $d_e = 2d$ when $d = 2$ but $d_e = d$ on the Hartmann6 problem. Our results in Fig. 3 indicate that with $d = 6$ HeSBO will have a much higher chance of reaching an optimum with $d_e = 2d$, so we evaluate this alongside their original choice of $d_e = d$.

Fig. 4 shows optimization performance for three HDBO tasks: the Branin problem extended to $D$=100 as described above; the Hartmann6 problem extended to $D$=1000; and the Gramacy problem extended to $D$=100. The Gramacy problem (Gramacy et al., 2016) includes two black-box constraints. The linear embedding methods (ALEBO, REMBO, and HeSBO) can naturally be extended to constrained optimization as described in Appendix A.5. The $D$=100 problems were repeated with 50 runs, and the $D$=1000 problem was repeated with 25 runs. Appendix A.6 provides additional details of the benchmark methods, additional experimental results (including plots of log regret and error bars), and an extended analysis of the results.

For all problems, ALEBO had the best average optimization performance. Relative to other linear embedding approaches, ALEBO also had low variance in the final best-value, which is important in real applications where one can typically only run one optimization run. For the $D$=1000 problem, REMBO-$\gamma k_\Psi$, EBO, and Add-GP-UCB did not finish a single run after 24 hours and so were terminated and not included in the results. These methods, along with SMAC and CMA-ES, also do not support blackbox constraints and so were not included in the results for the Gramacy problem.

We used the Branin problem to explore the sensitivity of optimization performance to $D$ and $d_e$, by varying $d_e$ from 2 to 8 and $D$ from 50 to 1000. We found that $d_e = d$ performed significantly worse than larger values, but for $d_e > d$ and across all values of $D$ there was little change in the BO performance. Figures with these results are in Appendix A.6.

## 7    POLICY SEARCH FOR ROBOT LOCOMOTION

Next, we evaluate our approach on a hexapod robot simulation for learning walking controllers. Sample efficiency is crucial in robotics as collecting data on real robots is time consuming and can cause wear-and-tear on the robot. We optimize the walking gait of the simulated hexapod robot "Daisy" (Hebi Robotics, 2019). The Daisy robot is simulated in PyBullet (Coumans & McCutchan, 2008), and has 6 legs with 3 motors in each leg, as shown in Fig. 5. The goal is to learn the policy parameters that enable the robot to walk to a target location while avoiding high joint velocities and height deviations. More details about this task can be found in Appendix A.7.

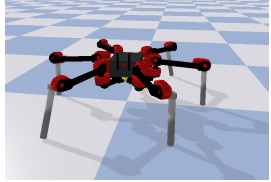

Figure 5: The simulated hexapod robot Daisy.

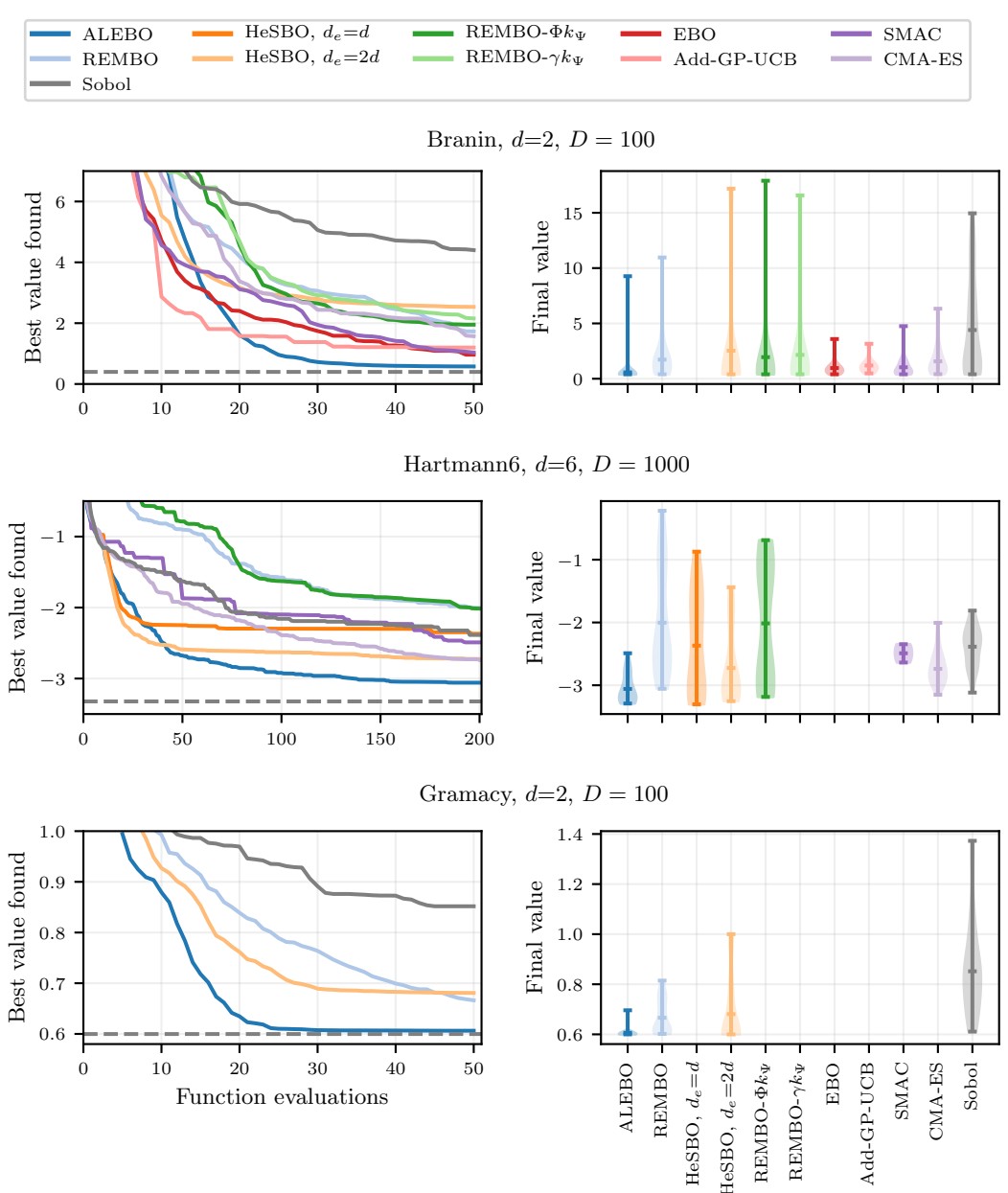

Figure 4: Optimization performance on three HDBO minimization problems. For each row, the left plot shows the best value by each iteration, averaged over repeated runs. The right plot shows the distribution of the best value at the final iteration. For all three tasks, ALEBO achieved the best average performance, and had the lowest variance in final performance of the linear embedding methods.

We use a Central Pattern Generator (CPG) (Crespi & Ijspeert, 2008) with $D = 72$ to control the robot. The CPG controller induces a cyclical motion in each joint of the robot. Different parameters of the CPG change the phase, amplitude, frequency, and offset of each joint. While the 72-dimensional controller assumes that each joint is independent of the others, one could construct a lower-dimensional embedding by coupling multiple joints. For example, the tripod gait in hexapods assumes three sets of legs synced, and out of phase with the remaining three legs. The dimensionality of the CPG controller can be reduced to 11 dimensions by restricting the movement to a tripod gait, and learning the common amplitude, offset and frequency of the joints. The existence of such

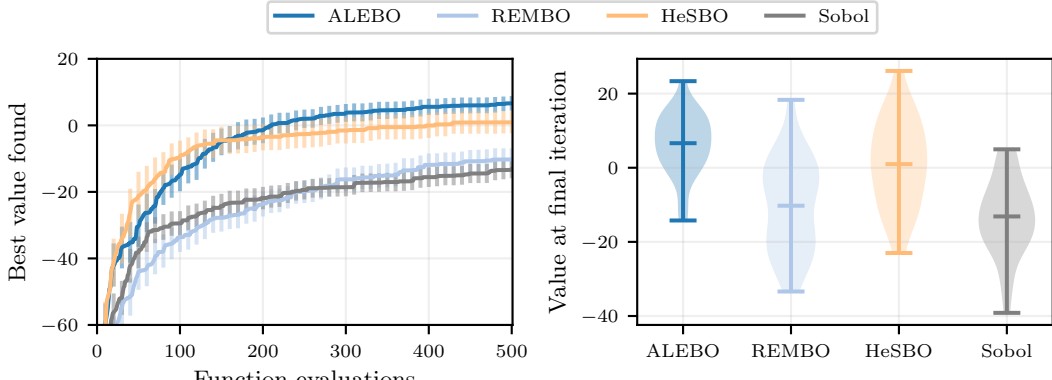

Figure 6: Optimization performance on the $D = 72$ hexapod locomotion task (higher is better). *(Left)* Mean and two standard errors (over 50 repeated runs) of the best value found by each iteration. *(Right)* Distribution of the best value found across repeated runs. ALEBO had the best average performance, and the lowest variance.

low-dimensional parameterizations motivates the use of ALEBO for learning the parameters of the CPG controller, although it is not known if there is a *linear* low-dimensional representation. In a real robot, each motor can have different physical properties, such as friction, damping, etc. This could make a pre-defined constrained space sub-optimal, and we could benefit from learning with a flexible embedding, as in ALEBO.

Fig. 6 shows optimization performance for the linear embedding methods on this task, which is a maximization problem. ALEBO improves on the state-of-the-art, with both higher mean performance and a lower variance (thus, a lower chance of poor performance). Expert tuning can achieve reward values above 40, so while ALEBO is an advance in terms of linear embedding BO, there is still much room for additional work in high-dimensional BO.

## 8    DISCUSSION

Our work highlights the importance of two basic requirements for an embedding to be useful for optimization that are often not examined critically by the literature: 1) the function must be well-modeled on the embedding; and 2) the embedding should contain an optimum. To the first point, we showed how polytope constraints on the embedding eliminate boundary distortions, and we derived a Mahalanobis kernel appropriate for GP modeling in a linear embedding. These two contributions allow effective modeling in the embedding space. To the second point, we developed an approach for computing the probability that the embedding contains an optimum, which we then used to construct embeddings with a higher chance of containing an optimum, via hypersphere sampling and selecting $d_e$ larger than $d$.

These same two considerations are important for any embedding, not just linear. For instance, when constructing a VAE for BO it will be equally important to ensure the function remains well-modeled on the embedding and that box bounds are not handled in a way that adds distortion. We must also ensure that the VAE embedding captures enough of the ambient space to have a high chance of containing an optimum. With linear embeddings we were able to derive analytical quantities for answering these questions—more work in this area is needed for nonlinear embeddings. Here we applied linear constraints to restrict the acquisition function optimization to points that project up inside the ambient box bounds. For a VAE these constraints will be general nonlinear functions, but their gradients can be backpropped and so constrained optimization could be done in a similar way.

Given $D$ and $d$, we can solve (2) to determine the probability of containing an optimum for any $d_e$, and thus select $d_e$ based on a desired target probability. We showed on test problems that BO performance was not too sensitive to the exact choice of $d_e$. In reality, such as in the robot locomotion task, we do not know $d$, or even if the problem has low-dimensional linear structure. In this case selecting an appropriate embedding dimension remains an important open question.

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

# A  APPENDIX

This appendix contains a number of additional results and analyses to supplement the main text.

## A.1  HeSBO Embeddings

We consider HeSBO embeddings in the case of a random axis-aligned true subspace, and a uniform prior on the location of the optimum within that subspace. As explained in Sec. 4, with $d = 2$ and this prior, regardless of $d_e$ or $D$ there are three possible embeddings: (1) each of the active parameters are captured by a parameter in the embedding; (2) the embedding is constrained to the diagonal $x_{i_1} = x_{i_2}$; or (3) the embedding is constrained to the diagonal $x_{i_1} = -x_{i_2}$. Fig. 7 shows these three embeddings for the Branin problem from the top row of Fig. 1.

Within the first embedding, the optimal value of 0.398 can be reached. Within the second, the best value is 0.925 and within the third it is 17.18. Under a uniform prior on the location of the optimum within a random axis-aligned true subspace, it is easy to compute the probability that the HeSBO embedding contains an optimum:

$$P_{\text{opt}}(d_e) = \frac{d_e!}{(d_e - d)! d_e^d}. \tag{3}$$

For $d = 2$, this is exactly the probability of the first embedding shown in Fig. 7. This probability increases with $d_e$, and is exactly the probability shown in Fig. 3.

## A.2  The Mahalanobis Kernel

When fitting the Mahalanobis kernel derived in Proposition 1, we use an approximate Bayesian treatment of $\Gamma$ to improve model performance while still maintaining tractability. We propagate uncertainty in $\Gamma$ into the GP posterior by first constructing a posterior for $\Gamma$ using a Laplace approximation with a diagonal Hessian, and then drawing $m$ samples from that posterior. The marginal posterior for $f(\boldsymbol{y})$ can then be approximated as:

$$p(f(\boldsymbol{y})) \approx \frac{1}{m} \sum_{i=1}^{m} p(f(\boldsymbol{y})|\Gamma^i).$$

Because of the GP prior, each conditional posterior $p(f(\boldsymbol{y})|\Gamma^i)$ is a normal distribution with known mean $\mu_i$ and variance $\sigma_i^2$. Thus the posterior $p(f(\boldsymbol{y}))$ is a mixture of Gaussians, which we can approximate using moment matching:

$$p(f(\boldsymbol{y})) \approx \mathcal{N}\left(\frac{1}{m} \sum_{i=1}^{m} \mu_i, \frac{1}{m} \sum_{i=1}^{m} \sigma_i^2 + \text{Var}_i[\mu_i]\right).$$

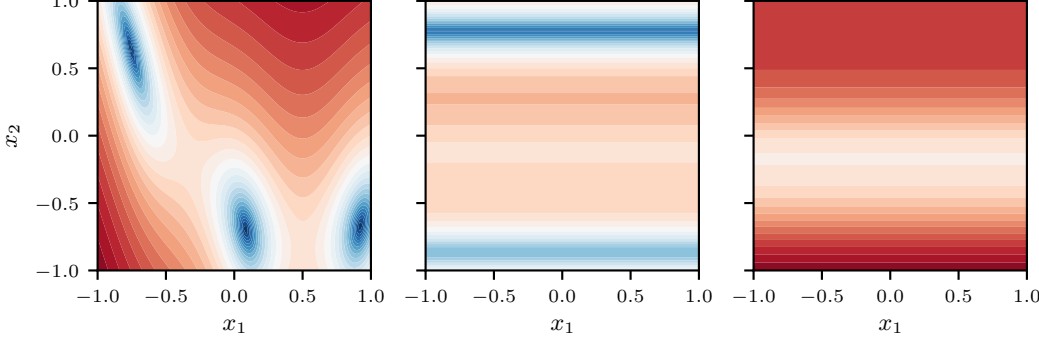

Figure 7: Three possible HeSBO embeddings of the $d = 2$ Branin function. *(Left)* The first embedding fully captures the function, and thus captures all three optima. *(Middle)* The second is restricted to the subspace $x_1 = -x_2$. This subspace does not contain an optimum, but comes fairly close. *(Right)* The third embedding is restricted to the subspace $x_1 = x_2$ and does not come close to any optima.

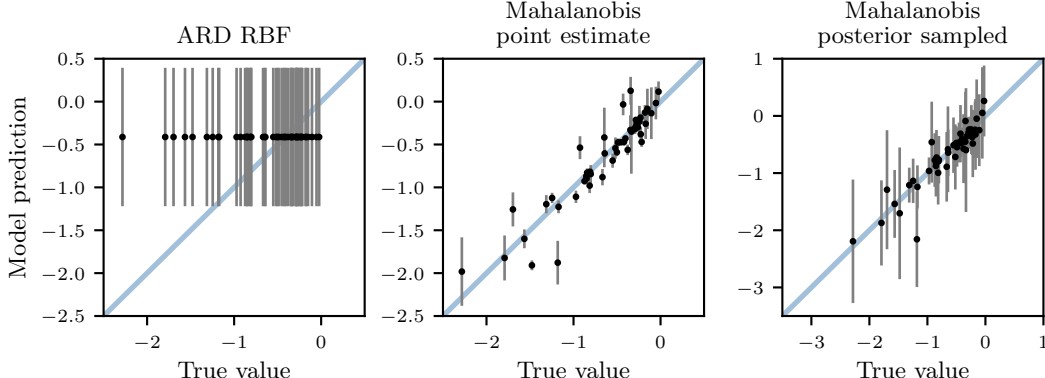

Figure 8: Test-set model predictions for three GP kernels on the same train/test data generated by evaluating the Hartmann6 $D$=100 function on a fixed linear embedding. A typical ARD kernel fails to learn and predicts the mean. The Mahalanobis kernel predicts well, and posterior sampling is important for getting reasonable predictive variance.

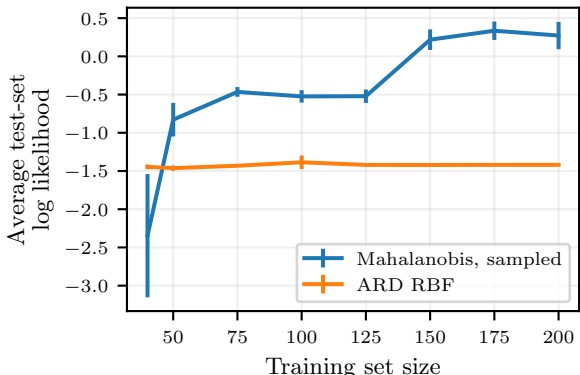

Figure 9: Average test-set log likelihood as a function of training set size, for training sets randomly sampled from a fixed linear embedding. Log marginal probabilities were averaged over a fixed test set of 1000 random points. For each training set size, 20 random training sets were drawn of that size and the figure shows the average result over those draws (with error bars for two standard errors). The ARD RBF kernel continues to predict the mean as the training set size is increased, while the Mahalanobis kernel is able to learn as the training set is expanded.

We do this to maintain a Gaussian posterior, under which acquisition functions like EI have analytic form and can easily be optimized, even subject to constraints as in (1).

We show the importance of the Mahalanobis kernel using models fit to data from the Hartmann6 $D$=100 function, from Fig. 1. We generated a projection matrix $B$ using hypersphere sampling to define a 6-d linear embedding. We then generated a training set (100 points) and a test set (50 points) within that embedding (that is, within the polytope given by (1)) using rejection sampling. We fit three GP models with different kernels to the training set, and then evaluated each on the test set: a typical ARD RBF kernel in 6 dimensions, the Mahalanobis kernel using a point estimate for $\Gamma$, and the Mahalanobis kernel with posterior marginalization for $\Gamma$ as described above.

Fig. 8 compares model predictions for each of these models with the actual test-set outcomes. With an ARD RBF kernel, the GP predicts the function mean everywhere, which is typical behavior of a GP that has failed to learn the function. With the same training data, the Mahalanobis kernel is able to make accurate predictions on the test set. Using a point estimate for $\Gamma$ significantly underestimates the predictive variance, which is rectified by using posterior sampling as described above. In BO exploration is driven by model uncertainty, so well-calibrated uncertainty intervals are especially important.

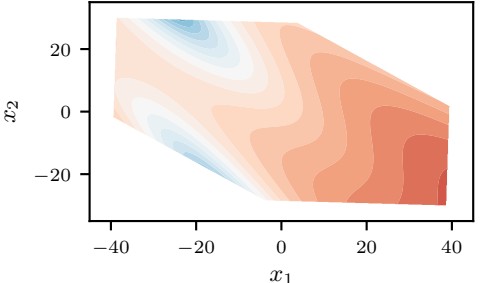 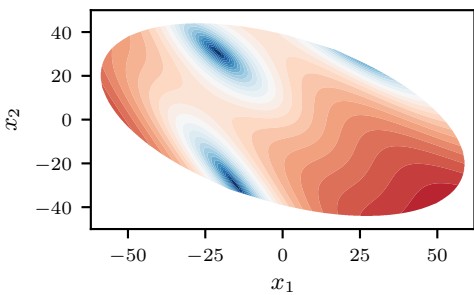

Figure 10: *(Left)* An embedding from a $\mathcal{N}(0,1)$ projection matrix on the same Branin $D = 100$ problem from Fig. 1 subject to constraints of (1). *(Right)* The embedding from the same projection matrix after normalizing the columns to produce unit circle samples. Sampling from the unit circle increases the probability that an optimum will fall within the embedding, and polytope bounds avoid nonlinear distortions.

Fig. 9 evaluates the predictive log marginal probabilities for the ARD RBF kernel and the Mahalanobis kernel with posterior sampling across a wide range of training sets with different sizes (without posterior sampling, Fig. 8 shows that the Mahalanobis point estimate significantly under covers and so has very poor predictive log marginal probabilities). We used the same linear embedding and Hartmann6 $D$=100 function used in Fig. 8 to sample 1000 test points which were held fixed. For each of 8 training set sizes ranging from 40 to 200, we randomly sampled 20 training sets from the embedding. For each training set, we fit the two GPs, made predictions on the 1000 test points, and then computed the average marginal log probability of the true values. Fig. 9 shows that as we vary the training set size from 40 to 200, the ARD RBF kernel continues to predict the mean, as in Fig. 8; even 200 points in the 6-d embedding are not sufficient to learn. For small training set sizes, the Mahalanobis kernel (with sampling) has high variance in log likelihood, as it has the potential to overfit and thus under cover. But for training set sizes of 50 and greater it had better predictive log likelihood than the ARD RBF, and continued to learn as the training set size was increased. For small datasets, the Mahalanobis kernel can overfit and thus have poor predictive likelihood, but for the purposes of BO, overfitting can be better than not fitting at all (predicting the mean), even when predicting the mean has better predictive log likelihood. This can be seen in the optimization results (Figs. 4 and 12) where ALEBO shows strong performance even with less than 50 iterations.

### A.3 POLYTOPE BOUNDS ON THE EMBEDDING

Rather than using projections to the box bounds $\mathcal{B}$, we specify polytope constraints in (1). Fig. 10 illustrates the embedding with these constraints for the same Branin $D = 100$ problem from the top row of Fig. 1. The embedding in the left figure was created with the REMBO strategy of sampling each entry from $\mathcal{N}(0,1)$. For the embedding in the right figure, that same projection matrix had each column normalized. This converts the projection matrix to be a sample from the unit circle, as described in Sec. 4.

The $\mathcal{N}(0,1)$ embedding does not contain any optima within the polytope bounds. Converting that projection matrix to a hypersphere sample rounds out the vertices of the polytope and expands the space to capture two of the optima. Consistent with Fig. 3, we see that hypersphere sampling significantly improves the chances of the embedding containing an optimum. Fig. 10 also shows that with the polytope bounds, we avoid the nonlinear distortions seen in Fig. 1.

### A.4 EVALUATING THE PROBABILITY THE EMBEDDING CONTAINS AN OPTIMUM

As in other parts of the paper, we consider a uniform prior on the location of the optimum within a random axis-aligned subspace. A random true projection matrix $\boldsymbol{T}$ is sampled by selecting $d$ columns at random and setting each to one of the $d$-dimensional unit vectors. $\boldsymbol{z}^*$ is then sampled uniformly at random from $[-1, 1]^d$. $\boldsymbol{B}$ is sampled according to the desired strategy, which in our experiments was REMBO, HeSBO, or hypersphere. Given these three quantities, we can evaluate whether or not $\boldsymbol{B}$ contains an optimum subject to the constraints of (1) by solving the following

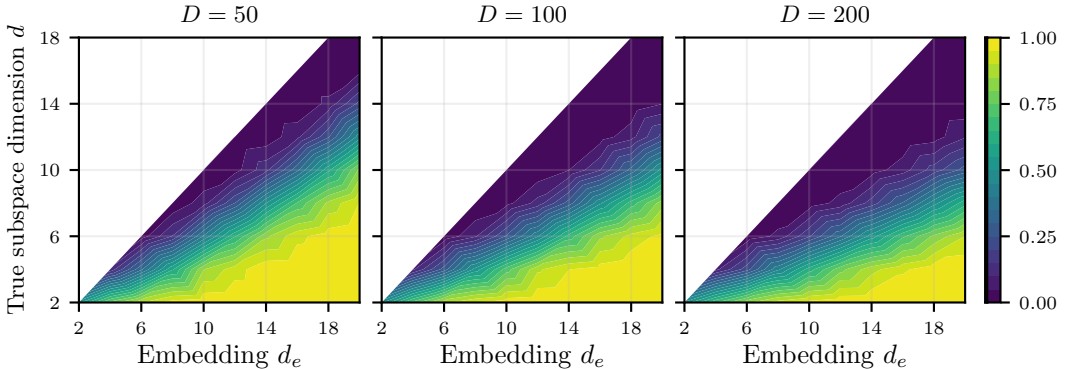

Figure 11: $P_{\text{opt}}$ for hypersphere sampling, as estimated in Fig. 3 but here for a wider range of values of $d$ and $D$. Contour color indicates $P_{\text{opt}}$. Doubling $D$ decreases $P_{\text{opt}}$ for $d$ and $d_e$ fixed, however even at $D = 200$, high values of $P_{\text{opt}}$ with reasonable values of $d_e$ can be had for many values of $d$.

linear program:

$$\begin{aligned}
\text{maximize } & \mathbf{0}^\top \boldsymbol{x} \\
\text{subject to } & \boldsymbol{T}\boldsymbol{x} = \boldsymbol{z}^*, \\
& (\boldsymbol{B}^\dagger \boldsymbol{B} - \boldsymbol{I})\boldsymbol{x} = \mathbf{0}, \\
& \boldsymbol{x} \geq -\mathbf{1}, \\
& \boldsymbol{x} \leq \mathbf{1}.
\end{aligned}$$

If this problem is feasible, then the embedding produced by $\boldsymbol{B}$ contains an optimum. If it is infeasible, then it does not. Solving this over many draws of $\boldsymbol{T}$, $\boldsymbol{z}^*$, and $\boldsymbol{B}$ produces an estimate of $P_{\text{opt}}$ under that prior for the location of optima. Here we used a uniform prior, but this linear program can be taken to compute $P_{\text{opt}}$ under any prior.

Fig. 11 shows $P_{\text{opt}}$ for a wide range of values of $d$ and $D$, for hypersphere sampling. Across this wide range we see that for many values of $d$ we can achieve high values of $P_{\text{opt}}$ with reasonable values of $d_e$, even for relatively high values of $D$.

### A.5 HANDLING BLACK-BOX CONSTRAINTS IN HIGH-DIMENSIONAL BAYESIAN OPTIMIZATION

In many applications of BO, in addition to the black-box objective $f$ there are black-box constraints $c_j$ and we seek to solve the optimization problem

$$\begin{aligned}
\text{minimize } & f(\boldsymbol{x}) \\
\text{subject to } & c_j(\boldsymbol{x}) \leq 0, \quad j = 1, \ldots, J, \\
& \boldsymbol{x} \in \mathcal{B}.
\end{aligned}$$

In most settings the constraint functions $c_j$ are evaluated simultaneously with the objective $f$. Constraints are typically handled in BO by fitting a separate GP to each outcome (that is, to $f$ and to each $c_j$). The acquisition function is then modified to consider not only the objective value but also whether the constraints are likely to be satisfied (*e.g.*, Gardner et al., 2014).

The extension of BO in an embedding to constrained BO is straightforward, so long as the same embedding is used for every outcome. A separate GP (in our case, using the Mahalanobis kernel) is fit to data from each outcome. Because the embedding is shared, predictions can be made for all of the outcomes at any point in the embedding. This allows us to evaluate and optimize an acquisition function for constrained BO in the embedding. Once a point is selected, it is projected up to the ambient space and evaluated on $f$ and each $c_j$ as usual. Random projections are especially well-suited for constrained BO because there is no harm in requiring the same projection for all outcomes, since it is a random projection anyway.

A.6    ADDITIONAL EXPERIMENTAL RESULTS

Here we provide results from an additional problem (Hartmann6 $D$=100), three additional methods (LineBO variants), and provide a study of the sensitivity of ALEBO performance to $d_e$ and $D$. We also provide implementation details for the experiments.

A.6.1    METHOD IMPLEMENTATIONS AND EXPERIMENT SETUP

The linear embedding methods (REMBO, HeSBO, and ALEBO) were all implemented using BoTorch, a framework for BO in PyTorch (Balandat et al., 2019), and so used the same acquisition functions and the same tooling for optimizing the acquisition function. EI was the acquisition function for the Hartmann6 and Branin benchmarks, and NEI (Letham et al., 2019) was used to handle the constraints in the Gramacy problem. ALEBO and HeSBO were given a quasirandom initialization of 10 points from a scrambled Sobol sequence. REMBO was given a Sobol initialization of 2 points for each of its 4 projections used within a run.

The remaining methods used reference implementations from their authors with default settings for the package: REMBO-$\phi k_\Psi$ and REMBO-$\gamma k_\Psi$[1]; EBO[2]; Add-GP-UCB [3]; SMAC[4]; CMA-ES[5]; and CoordinateLineBO, RandomLineBO, and DescentLineBO[6]. EBO requires an estimate of the best function value, and for each problem was given the true best function value. SMAC and CMA-ES require an initial point, and were given the point at the center of the ambient space box bounds.

The function evaluations for all problems were noiseless, so the stochasticity throughout the run and in the final value all comes from stochasticity in the methods themselves. For linear embedding methods the main sources of stochasticity are in generating the random projection matrix and in the quasirandom initialization.

A.6.2    ANALYSIS OF EXPERIMENTAL RESULTS

Fig. 12 provides a different view of the benchmark results of Fig. 4, showing log regret for each method, averaged over runs with error bars indicating two standard errors of the mean. This is evaluated by measuring the difference between the best point found so far, subtracting from that the optimal value for the problem, and then taking the log of that difference. The results are consistent with those seen in Fig. 4, and the standard errors show that ALEBO's improvement in average performance over the other methods is statistically significant. We now discuss some specific aspects of these experimental results.

**Branin $D$=100**    The additive kernel methods and SMAC all performed similarly on this problem, and, starting from around iteration 20, ALEBO performed the best. The distribution of final iteration values shows that in one iteration the ALEBO embedding did not contain an optimum and so achieved a final value near 10. However, across all 50 runs nearly all achieved a value very close to the optimum, leading to the best average performance.

The poor performance of HeSBO on this problem (particularly in Fig. 4 without the log, where it is outperformed by all methods other than Sobol) can be attributed entirely to the embedding not containing an optimum. Recall that for this problem there are exactly three possible HeSBO embeddings, which are shown in Fig. 7. As explained in Appendix A.1, the first embedding contains the optimum of 0.398, while the best value in the other embeddings are 0.925 and 17.18. Thus, if the BO were able to find the true optimum within each embedding with the budget of 50 function evaluations given in this experiment, the expected best value found by HeSBO would be:

$$0.398 P_{\text{opt}} + 0.925 \left( \frac{1 - P_{\text{opt}}}{2} \right) + 17.18 \left( \frac{1 - P_{\text{opt}}}{2} \right).$$

---

[1]github.com/mbinois/RRembo
[2]github.com/zi-w/Ensemble-Bayesian-Optimization
[3]github.com/dragonfly/dragonfly, with option acq="add_ucb"
[4]github.com/automl/SMAC3, SMAC4AC mode
[5]github.com/CMA-ES/pycma
[6]github.com/jkirschner42/LineBO

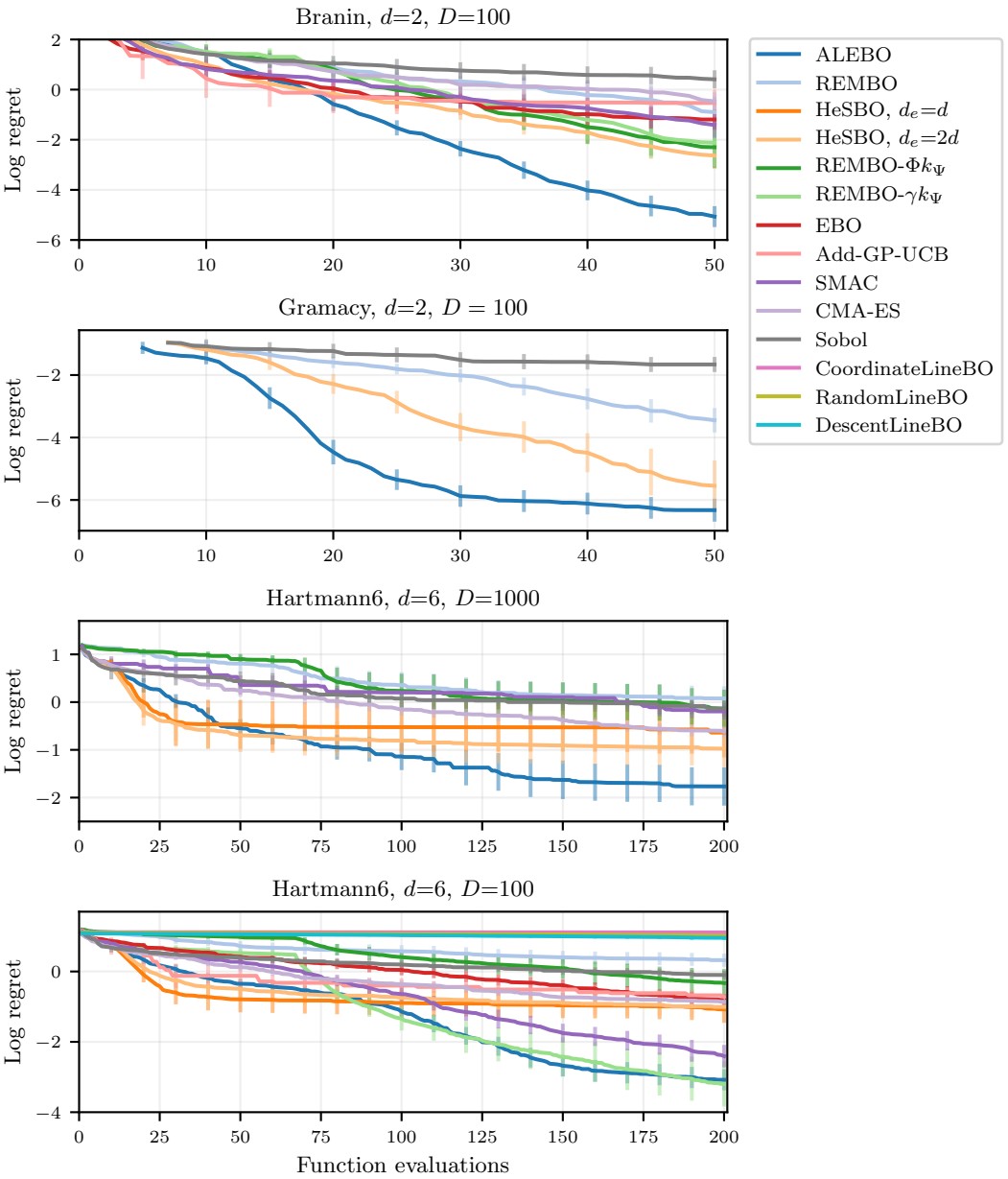

Figure 12: Log regret for the benchmark experiments of Fig. 4, plus Hartmann6 $D$=100. Each trace is the mean over repeated runs, with errors bars showing two standard errors of the mean. On the first three problems ALEBO performs significantly better than the other methods, and on Hartmann6 $D$=100 it is tied with REMBO-$\gamma k_\Psi$ as the best methods.

This is the best average performance one can hope to achieve using the HeSBO embedding on this problem. Using (3) we can compute $P_{\text{opt}}$ for $d_e = 4$ as 0.75, and it follows that the HeSBO expected best value is 2.56. This is nearly exactly the average best-value shown in Fig. 4. The poor performance of HeSBO is thus not related to BO, but comes entirely from the 12.5% chance of generating an embedding whose optimal value is 17.18. The presence of these embeddings can be clearly seen in the distribution of final best values in Fig. 4.

**Hartmann6 $D$=1000** As noted in the main text, the additive kernel methods and REMBO-$\gamma k_\Psi$ could not scale up to the 1000 dimensional problem. A nice property of linear embedding approaches is that the running time is not significantly impacted by the ambient dimensionality. Table

Table 1: Average running time per iteration in seconds on the Hartmann6 problem, $D$=100 and $D$=1000.

|  | $D$=100 | $D$=1000 |
|---|---|---|
| ALEBO | 29.5 | 32.6 |
| REMBO | 1.3 | 1.4 |
| HeSBO, $d_e$=$d$ | 1.0 | 1.6 |
| HeSBO, $d_e$=$2d$ | 1.0 | 1.4 |
| REMBO-$\phi k_\Psi$ | 2.1 | 1.1 |
| REMBO-$\gamma k_\Psi$ | 7.2 | — |
| EBO | 27.3 | — |
| Add-GP-UCB | 695.2 | — |
| SMAC | 9.5 | 404.5 |
| CMA-ES | 0.0 | 0.0 |
| Sobol | 0.0 | 0.0 |

1 gives the average running time per iteration for the various benchmark methods. Inferring the additional parameters in the Mahalanobis kernel and the added linear constraints make ALEBO slower than other linear embedding methods, but it has similar running time as EBO and is an order of magnitude faster than Add-GP-UCB, and at $D$=1000 is even an order of magnitude faster than SMAC. The average of 30s per iteration is short relative to the function evaluation time of typical resource-intensive BO applications.

**Hartmann6 $D$=100** REMBO performed worse than Sobol on this problem, despite there being a true linear subspace that satisfies the REMBO assumptions. The source of the poor performance is the poor representation of the function on the embedding illustrated in Fig. 1. The remaining methods all performed better than quasirandom. CMA-ES was competitive with all of the methods except SMAC, REMBO-$\gamma k_\Psi$, and ALEBO, which is somewhat surprising since it is not designed to have the same degree of sample efficiency as BO methods. HeSBO and Add-GP-UCB both did very well early on, but then got stuck and did not progress significantly after about iteration 50.

This problem was used to test three additional methods beyond those in Fig. 4: CoordinateLineBO, RandomLineBO, and DescentLineBO (Kirschner et al., 2019). These are recent methods developed for high-dimensional safe BO, in which one must optimize subject to safety constraints that certain bounds on the functions must not be violated. The performance of these methods can be seen in the bottom panel of Fig. 12: all three LineBO variants perform much worse than Sobol, and show almost no reduction of log regret. This finding is consistent with the results of Kirschner et al. (2019), who used the Hartmann6 $D$=20 problem as a benchmark problem. At $D$=20, they found that CoordinateLineBO required about 400 iterations to outperform random search, and even after 1200 iterations RandomLineBO and DescentLineBO did not perform better than random search. These methods are designed specifically for safe BO, which is a significantly harder problem than usual BO that has much worse scaling with dimensionality. The primary challenge for high-dimensional safe BO lies in optimizing the acquisition function, which is difficult even for relatively small numbers of parameters where there is no difficulty in optimizing the traditional BO acquisition function. The LineBO methods develop new techniques for acquisition function optimization, but do not consider difficulties with GP modeling in high dimensions, which is the main focus of HDBO work. LineBO methods perform very well on safe BO problems relative to other methods, but ultimately non-safe HDBO is not the problem that they were developed for, and so it is not surprising to see that they were not successful on this task.

### A.6.3 SENSITIVITY OF ALEBO TO EMBEDDING AND AMBIENT DIMENSIONS

We study sensitivity of ALEBO optimization performance to the embedding dimension $d_e$ and the ambient dimension $D$ using the Branin function. To test dependence on $d_e$, for $D = 100$ we ran 50 optimization runs for each of $d_e \in \{2, 3, 4, 5, 6, 7, 8\}$. To test dependence on $D$, for $d_e = 4$ we ran 50 optimization runs for each of $D \in \{50, 100, 200, 500, 1000\}$. Note that the $d_e = 4$ and $D = 100$ case in each of these is exactly the optimization problem of Fig. 4.

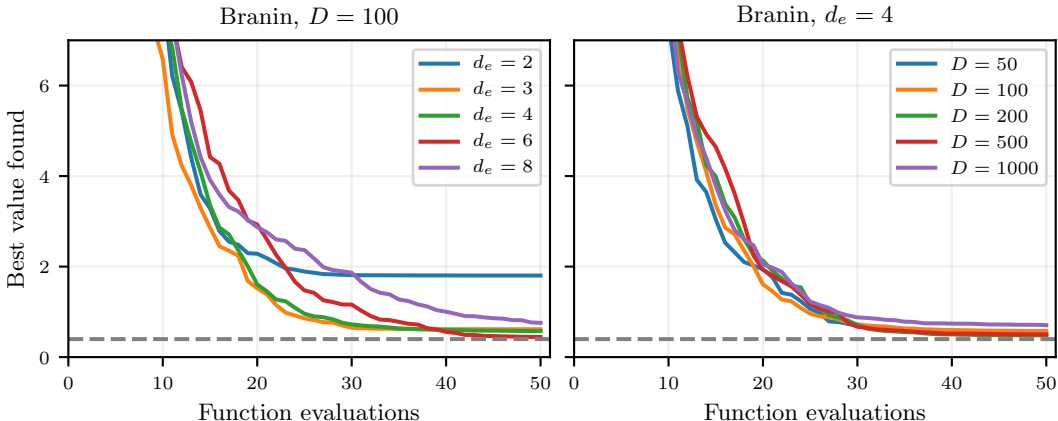

Figure 13: ALEBO performance on the Branin problem, (*Left*) as a function of embedding dimension $d_e$ and (*Right*) as a function of ambient dimension $D$. Performance shown is the average of 50 repeated runs. Optimization performance is poor with $d_e = 2$, but shows little sensitivity to $d_e$ for values greater than 2. Optimization performance shows little sensitivity with $D$, all the way up to $D = 1000$.

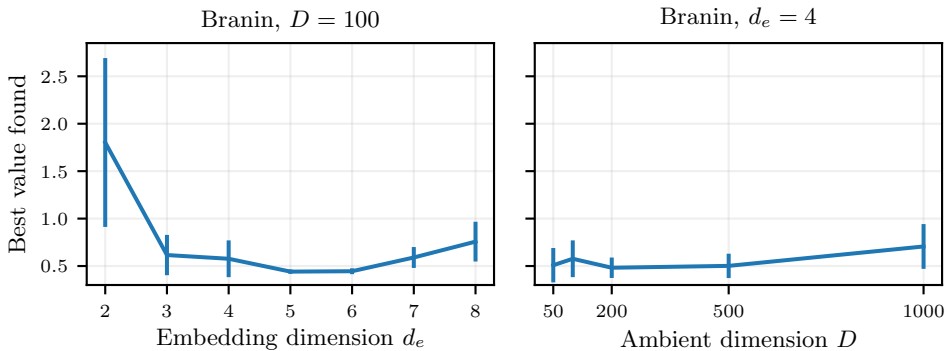

Figure 14: Final best value for the Branin problem optimizations Fig. 13, as mean with error bars showing two standard errors. With the exception of $d_e = 2$, optimization performance was good across a wide range of values of $d_e$ and $D$.

The results of the optimizations are shown in Figs. 13 and 14. For $d_e = d$, optimization performance was poor. From Fig. 3 we know this is because there is a low probability of the embedding containing an optimizer. Increasing $d_e$ increases that probability, but also increases the dimensionality of the embedding and thus reduces the sample efficiency of the BO in the embedding. This trade-off can be seen clearly in the figure: with $d_e = 2$ there is rapid improvement that then flattens out because of the lack of good solutions in the embedding, whereas for $d_e = 8$ the initial iterations are worse but then it ultimately is able to find much better solutions. Even at $d_e = 8$ the average best final value was better than that of any of the comparison methods in Fig. 4.

The ambient dimension $D$ will not directly impact the GP modeling in ALEBO, which depends only on $d_e$, however it will impact the probability the embedding contains an optimum as shown in Fig. 11. Consistent with the strong ALEBO performance for the Hartmann6 $D$=1000 problem, we see here that even increasing $D$ to 1000 produces only a small degradation in optimization performance. Even at $D = 1000$, ALEBO had better performance than the other benchmark methods had on $D = 100$.

### A.7 LOCOMOTION BENCHMARK PROBLEM

The task for the final set of experiments was to learn a gait policy for a simulated robot. As a controller, we use the Central Pattern Generator (CPG) from Crespi & Ijspeert (2008). The goal in

this task is for the robot to walk to a target location in a given amount of time, while reducing joint velocities, and average deviation from a desired height

$$f(\boldsymbol{p}) = C - ||\boldsymbol{x}_{\text{final}} - \boldsymbol{x}_{\text{goal}}|| - \sum_{t=0}^{T}(w_1||\dot{\boldsymbol{q}}_t|| - w_2|h_{\text{robot},t} - h_{\text{target}}|),  \tag{4}$$

where $C = 10$, $w_1 = 0.005$, and $w_2 = 0.01$ are constants. $\boldsymbol{x}_{\text{final}}$ is the location of the robot on a plane at the end of the episode, $\boldsymbol{x}_{\text{goal}}$ is the target location, $\dot{\boldsymbol{q}}_t$ are the joint velocities at time $t$ during the trajectory, $h_{\text{robot},t}$ is the height of the robot at time $t$, and $h_{\text{target}}$ is a target height. $T = 3000$ is the total length of the trajectory, leading to $30s$ of experiment. Cost is evaluated at the end of the trajectory.

