# OpenReview forum: "Re-Examining Linear Embeddings for High-dimensional Bayesian Optimization"
_ICLR.cc/2020/Conference — Reject_

### Official Review · AnonReviewer3 · 2019-10-22
**Official Blind Review #3**

**Rating:** 3

**Review:**

The authors investigate pitfalls common to random embedding-based approaches to high-dimensional Bayesian optimization (HDBO). Each of several practical shortcomings is separately analyzed and, subsequently, addressed in straightforward fashion:

  a. Large-scale distortions caused by clipping are handled by generalizing box constraints
     (in the embedded space) to the polytope corresponding to the set of points that project
     to the interior of the original search space.

  b. Local distortions caused by defining squared Euclidean distances in embedded spaces
     are handled by substitution for Mahalanobis distances.

  c. Embedded spaces potentially failing to contain optima are handled by constructing an
     estimator for the probability of this happening, which is then be used to pick better
     embeddings.


Feedback:
    To the extent that I enjoyed reading this piece, I am not sure that it warrants publication at this time. Specifically, the degree of novelty on offer seems minimal and the empirical results are underwhelming.

  1) Constraints on embedded candidate $\mathbf{y}$ are naively defined in terms of a polytope, but this formulation has previously been deemed impractical. If nothing else, it would be good to clarify this matter: why did preceding works chose not to explore this direction and/or how did you make it work here?

  2) Regarding use of Mahalanobis distances, evidence here (as provided in A.2) seems thin. Firstly, predictive MSE (Figure 6) seems like an odd choice of metric; log marginal probabilities would seemingly be more natural for GPs. Reporting of MSE is particularly suspect when results shown in Figure 7 indicate that the Mahalanobis distance based GPs are (markedly) overconfident. This issue is allegedly improved by marginalizing Mahalanobis parameters $\Gamma$; however, the appropriate baseline here would be ARD with marginalized lengthscales (which, to my knowledge, is not shown).

  3) Regarding ALEBO itself, Algorithm 1 states that acquisition functions were expressed in terms of an approximate posterior formed via moment matching against the Gaussian mixture formed by $m$ different samples of hyperparameters $\Gamma$?  One usually pushes this uncertainty through the acquisition function as, e.g., $\mathbb{E}_{\Gamma}[\alpha(\mathbf{x}; \Gamma)]$. What motivated this design choice?


Questions:
 - Does $\mathcal{B}$ need to be sampled or can it chosen to maximize $P_{opt}$?
 - Marginalization of hyperparameters
    - Did you jointly marginalize over all hyperparamerters or just Mahalanobis parameters $\Gamma$?
    - Why was a Laplace approximation used lieu of, e.g., slice sampling?

Nitpicks, Spelling, & Grammar:
  - Various figures: 'NewMethod' -> 'ALEBO'
  - Please report log immediate regret along with error bars
  - To the extent that testing on e.g. "high-dimensional" variants of Branin and Hartmann-6 is standard, it isn't particularly convincing.

**Experience Assessment:**

I have published one or two papers in this area.

**Review Assessment: Checking Correctness Of Derivations And Theory:**

I assessed the sensibility of the derivations and theory.

**Review Assessment: Checking Correctness Of Experiments:**

I assessed the sensibility of the experiments.

**Review Assessment: Thoroughness In Paper Reading:**

I read the paper at least twice and used my best judgement in assessing the paper.

---

> ### Author Response · Authors · 2019-11-15
> **Reply to Review 3**
>
> Thank you for your in depth review of the paper. We especially appreciate the thoughts around improving aspects of modeling and interesting extensions. Some of the concerns with the paper are around modeling decisions. We made clarifying edits in the paper, and address each question below. The primary concern is around the empirical analysis. This was admittedly weak, so in the revision it has been significantly expanded. We now evaluate 5 distinct problems, with D from 72 to 1000. We compare a total of 14 methods. We added a non-synthetic problem to our experimental results. We feel the breadth of our experiments now exceeds that typical of HDBO papers in top conferences. We have also added the specific empirical results requested below. We hope that with these improvements you will be able to improve your score. If there are additional concerns that prevent you from recommending acceptance, please let us know so we can make appropriate adjustments.
>
> 1) We are not exactly sure which previous work is being described. Binois et al. (2018) altered the REMBO projection to define a polytope that projects up to the same space as REMBO. However this was not directly incorporated into a constrained optimization problem. The reason (and perhaps the impracticality you reference) is possibly because acquisition function optimization used a genetic algorithm, which does not naturally support constraints. However, standard gradient optimizers  naturally support linear constraints with little additional cost (we use Scipy SLSQP). Table 1 shows there was only a 10% increase in ALEBO running time when increasing D from 100 to 1000, which increases the linear constraints from 200 to 2000. With gradients, there are no issues with handling thousands of linear constraints.
>
> 2) We originally selected MSE because it is typical for constructing GP learning curves. Per your request we have replaced that with the test-set log marginal prob. (Fig. 9), which did not alter the conclusions.
>
> To better clarify, we added a new plot directly showing GP predictions on a test set (Fig. 8). The ARD RBF kernel simply predicts the mean, while the Mahalanobis kernel makes good (as in, useful for BO) predictions.
>
> We are not aware of prior work on embeddings for HDBO that has shown GP fits within the embedding. We were surprised to discover that ARD RBF performs so poorly in the embedding, which led to the derivation of the appropriate kernel. The new figures and discussion in A.2 will make this more clear.
>
> ARD RBF with marginalization: Here we show the behavior of the kernel typically used for BO in a linear embedding. Based on Fig. 8 and the theoretical result in Prop. 1 that ARD kernels are not appropriate for this setting, we don’t expect hyperparameter marginalization to have much impact.
>
> 3) This was done to maintain Gaussian posteriors and thus analytic tractability of the acquisition function. This is a good idea and could be implemented using a fully MC-based acquisition while maintaining differentiability. We have prioritized improvements to exposition and additional experiments for this revision, but can add further discussion of this if needed.
>
> Choosing B: The idea of selecting B to maximize Popt is very interesting. The LP of A.4 will be an important tool for this optimization. This is an interesting area of research for HDBO, though beyond this paper.
>
> Why Laplace: We initially tried HMC, which was far too slow to be practical. We then turned to the Laplace approximation, which is computationally efficient and provides uncertainty estimates good enough for BO. The goal of our work here is to investigate the apparent lack of robustness in linear embedding-based approaches to BO and address the failure modes. Using slice sampling is a great idea, and would provide better uncertainty estimates than the Laplace approximation with less cost than HMC. We hope to test this in the future, though we note that even with the Laplace approximation ALEBO still consistently performs at least as well as the other HDBO methods.
>
> Log regret: We have added figures with log regret as requested (Fig. 12). We left “raw” values in the main text because in real BO tasks the value of a solution is typically the function value. The utility behind acquisition functions like EI and KG is the function value. Log regret allows for identifying small differences in performance that may not be practically significant. With log regret, the improvement of ALEBO over other methods appears even larger than in Fig. 4.
>
> Standard test functions: We agree it can be tiresome to see the same few benchmark tasks in every paper, though there is value in standard problems making it easier to compare and understand results across papers. To address these concerns, we have included a more realistic example in the robot locomotion task.
>
> Thank you again for taking the time to understand the paper at a technical level and for the quality of your feedback.

---

### Official Review · AnonReviewer1 · 2019-10-23
**Official Blind Review #1**

**Rating:** 1

**Review:**

This paper criticizes existing High Dimensional BO (HDBO) via linear embedding literature for the following reasons:

- Points in the embedded space projected mostly to the facet of the bounding box in the original space.
- The projection induces a distorted space which is not fit to be modeled by a GP.
- Linear projections do not preserve product kernels.
- Linear embeddings have low probability of containing an optimum.

The paper then proposes ALEBO which supposedly improves these aspects over other linear embedding BO techniques.

I have the following concerns regarding the authors' criticisms above:

1. What is wrong with the points being projected mostly to the facet of the original bounding box?
If I understand correctly, Theorem 3 of the REMBO paper proved that there exists y* \in R^{d_e} such that f(Ay*) = f(x*)
(i.e., the projected space contains the optimum) with high probability so to me, it does not really matter if the projection does not include the interior of the bounding box. Fig. 1 seems to make a point that most projections seem to indeed land on the facet of the bounding box (to be thorough, how many points did the authors sample to make this plot) but given what I said above, I do not think there is much of a point in Fig. 1 here.

2. The authors claimed that it is not appropriate to model the distorted space with GP, but ended up using GP
for ALEBO anyway (although with a different kernel). I understand that the authors did not project Ay back to B
like REMBO did, but the authors also gave me no reason to believe that this will improve things either. In fact, I think
the authors should show the space induced by ALEBO embedding as a comparison. I suspect that with the imposed
constraint -1 <= (B^T)y <= 1 the space will have discontinuous regions and is also not fit to be modeled
with any GP.

3. The authors stated that "A product kernel in the true subspace will not produce a product kernel in the embedding;
we will see this more explicitly in Sec. 5.1" but I did not see it explicitly in Sec. 5.1. At the very
least, I do not see how REMBO fails to do the same thing. Also, the authors claimed that "Inside the embedding,
moving along a single dimension will move across all dimensions of the true subspace, at rates depending
on the angles between the embedding and the true subspace". This seems like a very qualitative claim.
Can the authors formally define what this statement means, and prove it or at least provide some backup citations?

Other comments:

4. Why do the authors use conjugate transpose (if B^T means what i think it means) instead of normal transpose
when B is drawn from R^{d_e \times d}? Shouldn't they be the same?

5. Please explain the choice of \Epsilon(B) = {x: B^T B x = x}

6. The experiments provided are very limited. There is only one set of experiments showing performance of
ALEBO against other methods and it was done on a very small extrinsic dimension too (D = 100). I would like to see how
ALEBO scales with truly large dimension (REMBO also claimed that it could scale up to much higher extrinsic dimension). What about other important properties like does it guarantee that an optimum lies in the constraint space? What about rotational invariance? There are so many elements missing from the analysis.

Overall conclusion:

This paper is largely empirical and lacks technical depth. It is not at all convincing that the problem it
addresses is real, much less important. It also does not offer strong empirical evidence (too few experiments).
Given these reasons, I do not think the paper is not ready to be published as it is.

**Experience Assessment:**

I have published one or two papers in this area.

**Review Assessment: Checking Correctness Of Derivations And Theory:**

I assessed the sensibility of the derivations and theory.

**Review Assessment: Checking Correctness Of Experiments:**

I assessed the sensibility of the experiments.

**Review Assessment: Thoroughness In Paper Reading:**

I read the paper thoroughly.

---

> ### Author Response · Authors · 2019-11-15
> **Reply to Review 1**
>
> Thank you for your careful review of the paper. Points 1-5 primarily deal with aspects of the technical contribution of the paper that were unclear. We have updated the text to clarify these important points, and respond to each below. Point 6 deals largely with the empirical analysis. We now have 4 separate experiments in the main text that compare a total of 11 different methods for HDBO. These include a robot locomotion task. We also have 3 additional experiments in the supplement, with a total of 14 methods compared.
>
> The improvements in the exposition and clarity of the text should resolve points 1-5, and the greatly expanded experiments resolve point 6. With these points addressed, we hope you will be able to increase your rating for the paper; please let us know of any additional issues that prevent you from recommending acceptance of the paper.
>
> 1. Thm. 3 of the REMBO paper states that there exists an optimum in the embedding. However, there is no guarantee that that optimum will project up inside the box bounds! This means that if you are restricted to evaluating the function within the bounds (which is generally their purpose), REMBO has *no guarantee* that an optimum can be found in the embedding. We have better clarified this point by discussing it in Sec. 5.3. Why it matters that most points project to facets: Sec. 4 shows that points projected to the facet undergo a nonlinear projection that introduces kernel non-stationarity, an issue for GPs. The fact that nearly the entire volume of the space undergoes this nonlinear projection shows the severity of the issue. We have clarified this in the paper by re-ordering the first two points of Sec. 4, so we first describe the issue of nonlinear projections and then provide these results to indicate the extent of the issue. Number of samples is 1000, now in the text.
>
> 2. The reason the GP is inappropriate is because of the nonlinear projection. ALEBO restricts BO to the portion of the embedding where the projection is linear (no clipping), allowing for GP modeling. We now discuss this in Sec. 5.2 to make it more clear. The paper actually had a visualization of the ALEBO embedding, in the appendix due to space constraints; it is Fig. 10. For the constrained space, the constraints form a convex, bounded polytope; since it is convex, we can be sure there are no discontinuities. We have clarified this by expanding the discussion in Section 5.2.
>
> 3. We added to the text of that section (Sec. 4, “Linear projections...”) to describe this mathematically. We have also changed the reference to directly point to Prop. 1, which shows mathematically that a product kernel in the true space (ARD RBF) produces a not-product-kernel (Mahalanobis) in the embedding.
>
> 4. The dagger denotes matrix pseudoinverse; we now state this in Sec. 5.
>
> 5. We expanded Sec. 5.3 to derive this result.
>
> 6. We have greatly expanded the set of empirical experiments. We promoted two experiments from the appendix to the main text so there are now benchmark results shown for three synthetic problems. One of these has D=1000, as requested. Another includes black-box constraints, so they constitute a diverse set of problems. We also added experiments for a 72-D robot locomotion task.
>
> Guarantees of an optimum in the constraint space: let us re-emphasize that no linear embedding method has any guarantee of the embedding containing an optimizer when evaluations are restricted to box bounds. Thm 3 of the REMBO paper does not hold once you clip to box bounds. The probability the embedding contains an optimum will depend on the problem: D, d_e, and where the optimizer is in the embedding. In our paper we derive an unbiased estimator for this probability, under a prior for the location of the embedding (Sec. 5.3).
>
> All linear embedding methods inherently have rotational invariance: rotate the true subspace T to RT, and then rotate the embedding B to RB to produce the same problem.
>
> Overall conclusion: In points 1-5 there were many aspects of the technical contributions of this paper that were not clear. We have added additional text to address all of these points, which we expect will help readers to understand these subtle technical issues.
>
> The problem of HDBO is real and important, as it enables sample-efficient black-box optimization of any type of system. Many optimization problems involve more parameters than can be handled by a standard GP, and if work published in ICLR is any indication, it is plausible to expect that the underlying process can be represented within a low-dimensional subspace. We have also added a real HDBO problem to the paper, optimizing a 72-D controller for robot locomotion. These are real problems and the work we do in this paper produces a real advance in both technical understanding of the problem and in HDBO optimization performance.
>
> We expect you will find the expanded experiment section as described above to be more compelling.
>
> Thank you again for your review!

---

### Official Review · AnonReviewer2 · 2019-10-24
**Official Blind Review #2**

**Rating:** 3

**Review:**

This is a well-written paper and I enjoyed reading it. In summary the paper tries to address the following shortcomings of REMBO:

(1)	REMBO uses a random embedding to project a point in high dimensional space to a lower dimensional embedding space basing on the relation f(x) = f(Ay) with high probability, where x in D dimensions and y in d dimensions and d<<D. The problem of REMBO is that the relation f(x) = f(Ay) is only guaranteed with high probability and so, the embedding space may not contain an optimum. Second, When a point y  where Ay is outside the search space X, REMBO uses a projection to map Ay to its nearest point in X. This projection is not enough good. These observation are identified in paper of Binois et al (2018)as well.
(2)HESBO is an extension of REMBO that avoid above restrictions by proposing a new random projection. However, as the paper mentioned, HeSBO have a limitation is that the probability that the embedding will contain an optimum can be quite low!
(3)Besides, the paper also identify a new observation that linear projections do not preserve product kernels.

Then, the paper proposes a new solution of BO to overcome these restrictions by using a Mahalanobis kernel to avoid (3). This kernel is a replace of ARD Euclidean distance to a Mahalanobis distance. To avoid (1), the paper use equation 1 (please find in the paper). To avoid (2), they use the projection P_opt. In all , I think the theoretical contribution is good enough.

Having said that, I am a bit disappointed that this paper does not talk about LineBO (ICML 2019).  LineBO is a good solution for high dimensions without any assumption on structure like low effective dimensionality. It uses even one-dimensional subspaces to solve high dimensional problem with the strong theoretical guarantee. It do not need to learn subspace, and so it avoids disvantages (1), (2) and (3) that cause due to the fact that the embedding may not contain an optimum as mentioned above. Thus, LineBO is a stronger contender to the proposed algorithm. I will lift my rating if the author provide their response to this point.

Additionally, the author should compare their method to the algorithm of Binois et al (2018) that solved very well disadvantages of REMBO by setting bounds to avoid (1). Moreover, because the problem of the paper is high-dimensional Bayesian optimization under the assumption of low effective dimensionality, they should compare to other strong algorithms under the same assumption such as SI-BO algorithm( NIPS 2013) that used active learning to learn the low-dimensional subspace instead of using random embedding like REMBO.







**Experience Assessment:**

I have published one or two papers in this area.

**Review Assessment: Checking Correctness Of Derivations And Theory:**

I carefully checked the derivations and theory.

**Review Assessment: Checking Correctness Of Experiments:**

I assessed the sensibility of the experiments.

**Review Assessment: Thoroughness In Paper Reading:**

I read the paper thoroughly.

---

> ### Author Response · Authors · 2019-11-15
> **Reply to Review 2**
>
> Thank you for your review of our paper, and in particular for recommending some additional methods to include in the benchmark experiments. Per your request, we have included LineBO (3 variants), the method of Binois et al. (2018), and also the method of Binois et al. (2015). The results of these methods can be seen in Figs. 4 and 11, and we give some discussion of them in response to your comments below. We hope that as a result of these improvements you will be able to raise your rating as indicated. If there are any additional items that prevent you from recommending acceptance, please let us know as we continue improving the paper.
>
> ###
> “Having said that, I am a bit disappointed that this paper does not talk about LineBO (ICML 2019). LineBO is a good solution for high dimensions without any assumption on structure like low effective dimensionality. […] Thus, LineBO is a stronger contender to the proposed algorithm. I will lift my rating if the author provide their response to this point.”
>
> Thank you for pointing out the absence of LineBO in our discussion of HDBO methods. The LineBO paper primarily addresses the issue of performing “safe” BO in more than 1-2 dimensions: while previous methods (i.e., SafeOpt) involved discretizing the input space, LineBO is able to move along 1D subspaces within the search space.  In this sense, LineBO solves the problem of *high dimensional safe bayesian optimization*. However, the underlying probabilistic model in LineBO is still a standard GP, and so it does not help alleviate issues with estimating GPs with >20 dimensions, which is the subject of our work. Since they are focused on Safe BO, the LineBO methods are not necessarily appropriate for regular (non-safe) HDBO, and in fact most of the LineBO paper primarily focuses on problems that would not be considered high-dimensional for regular BO (but are for Safe BO). For instance the highest dimensional problem is only 24D; compare to 100D, the lowest in our paper.
>
> We added LineBO to our experiments per your request, and you can see the results in Fig. 12, in the appendix, for the Hartmann6 D=100 problem. LineBO performs very poorly on this task, and in fact performs much worse than random. This result is consistent with those in the LineBO paper: The LineBO paper uses Hartmann6 D=20 as a benchmark problem, and even at D=20 they show that CoordinateLineBO requires about 400 iterations to perform better than random search. With 1200 iterations RandomLineBO and DescentLineBO still did not perform better than random. Increasing the dimensionality from 20 to 100 can only further decrease performance.
>
> Ultimately, LineBO is meant for a different type of problem (Safe BO, not generic HDBO). We do not want to give readers the impression that LineBO is a bad method (since it is not, it is just not meant for this problem) so we included the LineBO comparison only in the appendix, where we give a paragraph to explain why the performance on this problem is as poor as it is (the last paragraph of A.6.2).
>
> ###
> “Additionally, the author should compare their method to the algorithm of Binois et al (2018) that solved very well disadvantages of REMBO by setting bounds to avoid (1). Moreover, because the problem of the paper is high-dimensional Bayesian optimization under the assumption of low effective dimensionality, they should compare to other strong algorithms under the same assumption such as SI-BO algorithm (NIPS 2013) that used active learning to learn the low-dimensional subspace instead of using random embedding like REMBO.”
>
> We have added a comparison to the algorithm of Binois et al. 2018 (REMBO-gamma k_Psi), as well as the algorithm of Binois et al. 2015 (REMBO-phi k_Psi) as requested. On the Branin problem these methods have similar performance as HeSBO and REMBO. On the Hartmann6 problem, REMBO-gamma k_Psi could not scale to D=1000, but for D=100 it was tied with ALEBO for the best-performing method.
>
> The Djolonga et al. 2013 SI-BO paper is very interesting and we do give a brief description of it in the related work. It uses an active subspace method, with low-rank matrix active learning methods to approximate gradients. The work is oriented around bandit problems with much larger budgets than typical BO problems. In their 100D benchmarks, over 500 points are used just for the random initialization, and subsequent optimization occurred over thousands, rather than hundreds of iterations. Djolonga et al. do benchmark experiments with the same 100D augmented Branin function (p17 of the extended version) that we used in Figs. 4 and 12. They use 2500 iterations, vs. 50 in our experiments. The simple regret reported in their figure is higher than that seen for ALEBO in Fig. 11. Ultimately, SI-BO is a very interesting method with advantages in learning the subspace as you note, but is meant for problems with a sample budget orders of magnitude larger.
>
> Thank you again for your review!

---

### Author Response · Authors · 2019-11-15
**Revised version**

We are pleased to post a revised version of the paper with several improvements to address the points raised by the reviewers. We have significantly expanded the empirical evaluation of the method. The revision includes an additional benchmark experiment, 5 new benchmark comparison methods, empirical experiments with a robotics application problem, and improved exposition to clarify areas as indicated by the reviews. Thank you for your reviews and for the improvements they have brought to the paper.

---

### Decision · Program_Chairs · 2019-12-19

**Decision:**

Reject

**Comment:**

This paper explores the practice of using lower-dimensional embeddings to perform Bayesian optimization on high dimensional problems.  The authors identify several issues with performing such an optimization on a lower-dimensional projection and propose solutions leading to better empirical performance of the optimization routine.  Overall the reviewers found the work well written and enjoyable.  However, the reviewers were concerned primarily about the connection to existing literature (R2) and the empirical analysis (R1, R3).  The authors claim that their method outperforms state-of-the-art on a range of problems but the reviewers did not feel there was sufficient empirical evidence to back up this claim.
 Unfortunately, as such the paper is not quite ready for publication.  The authors claim to have significantly expanded the experiments in the response period, however, which will likely make it much stronger for a future submission.